# Active Treatment Effect Estimation via Limited Samples

**Zhiheng Zhang**[1]  **Haoxiang Wang**[2]  **Haoxuan Li**[3]  **Zhouchen Lin**[4 5 6]

## Abstract

Designing experiments for causal effect estimation remains an enduring topic in both machine learning and statistics. While much of the existing statistical literature focuses on using central limit theorems to analyze asymptotic properties of estimators, a parallel line of research has emerged around theoretical tools that provide finite-sample error bounds, offering performance on par with—or superior to—the asymptotic approaches. These finite-sample results are especially relevant in active sampling settings where the sample size is limited (for instance, under privacy or cost constraints). In this paper, we develop a finite-sample estimator with sample complexity analysis and extend its applicability to social networks. Through simulations and real-world experiments, we show that our method achieves higher estimation accuracy with fewer samples than traditional estimators endowed with asymptotic normality and other estimators backed by finite-sample guarantees.

## 1. Introduction

Experimental design has long been regarded as the gold standard for causal effect estimation, providing a principled framework to establish causal relationships through controlled interventions. Building on the foundational work of Kirk (2009), researchers have developed a rich repertoire of methodologies—including randomized controlled trials (RCTs), regression-adjusted estimators, and instrumental variable techniques—that have been widely adopted across diverse disciplines such as medicine, economics, and the social sciences. Despite these advances, the fundamental challenge of how to design randomized experiments that estimate treatment effects both accurately and efficiently under constrained sample budgets remains underexplored.

This challenge is particularly pressing for two reasons. First, from a practical standpoint, data collection is often expensive, time-consuming, or ethically constrained. For example, in clinical trials, it is neither economically viable nor morally justifiable to administer a new drug uniformly across all patients to measure its effect, even when detailed patient profiles are available. Second, from a statistical perspective, analyzing the full dataset can sometimes degrade estimation quality due to overfitting and increased sensitivity to noise. A natural strategy to mitigate these issues involves active sampling, where a carefully chosen subset of individuals is selected to maximize information gain while minimizing sample complexity. However, existing methods often lack finite-sample guarantees, making their performance unpredictable in real-world, resource-limited settings.

Finite-sample guarantees are particularly crucial in active learning settings for two main reasons: (i) they provide explicit, non-asymptotic assurances on the accuracy of causal estimators for any given sample size, a property that is particularly valuable when data collection occurs sequentially under strict budget constraints, as emphasized by Ghadiri et al. (2024); and (ii) they offer concrete worst-case error bounds, ensuring robustness in high-stakes applications such as healthcare and policy evaluation, where even minor estimation errors can lead to significant real-world consequences. These considerations naturally lead to a central question: How can we achieve the most optimal finite-sample bounds possible for causal effect estimation within an active sampling framework?

Existing approaches provide initial insights into this problem. The seminal work of Efron (1971) analyzed the trade-off between covariate balance and robustness, laying the groundwork for later methods. Harshaw et al. (2024) extended this perspective by developing a Gram–Schmidt Walk-based (GSW) estimator, which was the first to establish finite-sample error bounds for average treatment effect (ATE) estimation in an experimental design context.

---

[1] School of Statistics and Data Science, Shanghai University of Finance and Economics, Shanghai 200433, P.R. China [2] School of Mathematical Sciences, Peking University [3] Center for Data Science, Peking University [4] State Key Lab of General AI, School of Intelligence Science and Technology, Peking University [5] Institute for Artificial Intelligence, Peking University [6] Pazhou Laboratory (Huangpu), Guangzhou, China. Correspondence to: Haoxuan Li <hxli@stu.pku.edu.cn>, Zhouchen Lin <zlin@pku.edu.cn>.

*Proceedings of the 42nd International Conference on Machine Learning*, Vancouver, Canada. PMLR 267, 2025. Copyright 2025 by the author(s).

However, their approach focuses on the full population and does not incorporate regression adjustment. More recently, Addanki et al. (2022); Ghadiri et al. (2024) leveraged leverage score sampling to derive finite-sample guarantees under active sampling. They demonstrated that applying leverage score-based sampling to treatment and control groups yields an $\epsilon$-approximation error with sample complexities of $\mathcal{O}(d \log(d) + d/\epsilon)$ and $\mathcal{O}(d \log(d) + m')$[1], respectively—both independent of the full sample size $n$.

While these results represent significant progress, existing methods do not necessarily achieve optimal dependence on the covariate dimension $d$. This gap is particularly critical in high-dimensional settings and network-dependent scenarios, where each individual's outcome may be influenced by treatments administered to their neighbors. Understanding the optimality of sample complexity is fundamental, as it establishes theoretical performance limits that can inform the design of more efficient estimation procedures.

To address these challenges, we introduce a novel active sampling strategy for treatment effect estimation that integrates partitioning and subsampling under explicit sample constraints. Our method, termed `RWAS`, achieves a sample complexity of only $O\left(\frac{d}{\varepsilon}\right)$, ensuring both efficiency and theoretical robustness. Unlike previous approaches, our method comes with finite-sample guarantees, making it well-suited for scenarios where large-scale experimentation is infeasible due to cost, ethical concerns, or logistical constraints. Furthermore, our framework naturally extends to network interference settings, addressing the additional complexities introduced by treatment spillover effects in networked environments. Empirical evaluations on synthetic and real-world datasets demonstrate that `RWAS` consistently outperforms existing baseline methods, achieving higher estimation accuracy with fewer samples. These findings establish our method as a theoretically sound and practically effective solution for sample-efficient causal inference. Our key contributions are as follows:

- We propose an efficient active sampling method, `RWAS`, with a sample complexity constrained by $O\left(d/\varepsilon\right)$, accompanied by finite-sample error guarantees.

- We demonstrate the (near) optimality of our sample size and extend the framework to handle network interference scenarios.

- We empirically validate our method on synthetic and real-world datasets, showing that it consistently outperforms existing baselines in terms of both accuracy and efficiency.

Our paper is organized as follows. Section 2 provides the

---

[1] $m'$ is a constant more minor than the full sample size.

literature overview of the finite-sample bound via active sampling and its variants. Section 3 introduces our framework. Section 4 presents our main result, along with two additional discussions upon lower bound (optimality) in Section 5.1 and network interference case in Section 6, respectively. Finally, our method performance is demonstrated by synthetic and real-world experiments in Section 7, and we end up with a conclusion and future work in Section 8. Anonymous code is available at `https://github.com/ZHzhang01/ICML_Finite_sample/settings`.

## 2. Literature review

In the section, we start from our initial setting, *Randomized experiment in causal inference*, to briefly review two main technical challenges in our paper-*finite-sample guarantee* and *active sampling*, along with the formulation of estimator: *Regression-based estimator*. To end up, we discuss the *causality estimation under network interference* to prepare for our extension in Section 6.

**Randomized experiment in causal inference.** Different from observational studies (Rosenbaum & Rubin, 1983; Wang et al., 2023; Zhang et al., 2023b; Zhang, 2024; Zhang & Su, 2024; Wang et al., 2025), randomization in experiments and optimization has been considered an important way to provide robustness and balance covariates in causal inference (Efron, 1971; Kallus, 2018; Zhang, 2022; Li et al., 2023a;b; Zhou et al., 2025), especially in finite samples (Bertsimas et al., 2015; Basse et al., 2023; Bai, 2023). For the first time, Harshaw et al. (2024) explicitly reveals the balance-robustness trade-off under the Gram-Schmidt design. Apart from that, rerandomization is also a proper way to balance covariates (Morgan & Rubin, 2015; Li & Ding, 2020), but time efficiency is an important issue to be considered (Johansson & Schultzberg, 2022).

**Finite-sample guarantee in experimental design.** The effect of finite sample size to the experimental design has been discussed for a long time (Kacewicz & Milanese, 1992; Way et al., 2010). However, to our knowledge, Harshaw et al. (2024) is the exploring paper to thoroughly induce the finite sample bound in experimental design. Stepping forward, Addanki et al. (2022); Ghadiri et al. (2024) extend the bound, taking account of the sampling procedure.

**Active sampling in experimental design.** As a form of importance sampling, active sampling proposed in Chen & Price (2019) selects samples through weights assigned to each sample point. Later work extends active sampling and linear regression to various settings, including online experiments (Fontaine et al., 2021) or other norm spaces Musco et al. (2022). In parallel, leverage score is another effective importance sampling method (Mahoney et al.,

2011; Drineas et al., 2012), employed in causal inference in Addanki et al. (2022); Ghadiri et al. (2024). See Tokdar & Kass (2010); Tabandeh et al. (2022) for reviews on importance sampling. Another remarkable type of method is to treat the problem as an active learning task (Jesson et al., 2021; Toth et al., 2022; Zhang et al., 2023a).

**Regression adjustment in causal inference.** Regression adjustment is a common way to improve efficiency in causal inference (Li & Ding, 2020; Bulbulia et al., 2021; Zhang et al., 2020). However, few works derive finite sample bound for causal estimators with regression adjustment, especially in finite population (Mou et al., 2022), except for Ghadiri et al. (2024) used as our baseline in the following text.

**Causal inference in networked data.** Conducting causal inference in networked data, also known as causal inference with interference, has been long aware of both statistically and experimentally (Hudgens & Halloran, 2008; Vander-Weele et al., 2012; Ogburn & VanderWeele, 2014; Su & Zhang, 2023; Zhang & Wang, 2024; Li et al., 2024; Yang et al., 2024). Exposure mapping is a classical way to model the neighborhood impact on individual outcome (Aronow & Samii, 2017; Leung, 2022). Although there has been some works on randomized tests such as A/B testing with interference (Basse & Airoldi, 2018; Basse et al., 2019), to our knowledge, there hasn't been work so far dealing with interference with limited samples, nor deriving finite sample bound for causal inference with interference.

## 3. Framework and notation

In this section, we define the basic notation of the article. For each individual $i$, we use $\boldsymbol{X}_i \in \mathbb{R}^d$ to denote the covariates of $i$-th individual. We represent the total sample information by $\boldsymbol{X}$, where $\boldsymbol{X} = [\boldsymbol{X}_1, \boldsymbol{X}_2, ... \boldsymbol{X}_n]^\top$ is an $n * d$ matrix. $\boldsymbol{X}_{i,j}$ denotes the $(i,j)$-th elements of matrix $\boldsymbol{X}$, namely, the $j$-th coordinate of $\boldsymbol{X}_i$. By performing a column-wise Gram-Schmidt orthogonalization on $\boldsymbol{X}$, we assume without loss of generality that the columns of $\boldsymbol{X}$ is a set of orthogonal basis in $\mathcal{R}^n$. Here $n$ represents the number of samples and $d$ represents the sample dimension. Moreover, $Y_i(1), Y_i(0)$ denotes the potential outcome on each individual $i$ when the treatment is chosen to be 0 or 1. The observed data $Y_i = \mathbf{1}(z_i = 1)Y_i(1) + \mathbf{1}(z_i = 0)Y_i(0)$. We denote $\boldsymbol{Y}^{(0)} := \{Y_i(0)\}_{i \in [n]}$ and $\boldsymbol{Y}^{(1)} := \{Y_i(1)\}_{i \in [n]}$. For each integer $k$, we use $[k]$ to replace $\{1, 2, ...k\}$. Moreover, we denote the treatment assignment as $z_i \in \{0, 1\}$ and $\boldsymbol{Z} := \{z_1, z_2, ...z_n\}$ for each individual.

**Assumption 3.1.** (SUTVA assumption) Each individual's treatment could not affect other individuals' outcomes (Imbens & Rubin, 2015).

**Definition 3.2.** (**I**ndividual **T**reatment **E**ffect) (ITE) The individual treatment effect (ITE) is denoted as the different

between the potential outcome when the treatment is chosen as 0 or 1. Namely, we denote the ITE of $i$-th individual as $t_i := Y_i(1) - Y_i(0)$. Moreover, $\boldsymbol{T} = \{t_0, t_1, ....t_n\}$ indicates the vector of individual treatment effect (ITE) which serves as the ground truth.

**Definition 3.3.** (**A**verage **T**reatment **E**ffect) (ATE) We denote the average treatment effect as $\tau := \frac{1}{n} \sum_{i=1}^n t_i = \frac{1}{n}(\sum_{i=1}^n Y_i(1) - Y_i(0))$.

Our question is: Given the covariate information of $n$ samples, which is denotes as $\boldsymbol{X} \in \boldsymbol{R}^{n*d}$, how can we accurately estimate the individual treatment effect ($\{t_i\}_{i=1}^n$) and average treatment effect ($\tau$), using as few sample points ($s \ll n$) as possible, with non-asymptotic guarantee?

## 4. Method

We achieve a new active sampling algorithm with finite-sample guarantees by adopting a superior beyond-leverage-score sampling approach. The implementation process consists of two critical steps: (i) Refinement of Design-Based Results: We first adapt the comprehensive theoretical results from Chen & Price (2019) within the design-based setting and formulate them into a novel (**I**terative **R**e-weighting in **D**esign-based Set- ting) IRD algorithm. This algorithm can effectively control fitting errors with high probability while requiring a significantly smaller sample complexity. (ii) Integration into Experimental Design: We then incorporate the IRD algorithm into the experimental design, leveraging the regression coefficients obtained from IRD to adjust classical finite-sample estimators, thereby enhancing the accuracy of finite-sample bounds. This structured approach allows us to develop a more efficient and theoretically grounded active sampling framework with strong finite-sample guarantees.

### 4.1. Theoretical results on active regression

We first consider the linear case and define the linear function family as $\mathcal{F}$, namely: $\mathcal{H}:=\{h : h(\boldsymbol{X}) = \boldsymbol{X}\boldsymbol{\beta}\}$, where $\boldsymbol{\beta}$ is an arbitrary $d \times 1$ vector. Our goal is to derive a reweighting strategy that distinguishes individuals who have a significant impact on the fitting process. This approach serves as a refinement of Chen & Price (2019), in which we transfer the strategy (Definition 5 in Chen & Price (2019)) to the design-based setting; namely, in the finite population, all randomness originates solely from the assignment process rather than from inherent data variability.

**Definition 4.1.** (**Good** $\varepsilon$-**reweighting sampling strategy, finite-sample version**) Consider a sampling procedure with $m(\le n)$ iterations. Let $\boldsymbol{\varepsilon} := (\varepsilon_1, \varepsilon_2, \varepsilon_3)$. In each iteration $i \in [m]$, we extract one sample whose index is $r(i) = j$ with probability $P_{ij}$, where $P_{ij}$ is a probability derived from Algorithm 2. We introduce the weighting coefficient

as $w_i = \alpha_i/P_{ir(i)}$, where $\alpha_i > 0$. We say it is a "good $\varepsilon$-reweighting sampling strategy" if (i) Defining a matrix $\boldsymbol{A}$ with elements $A(i,j) = \sqrt{w_i}\boldsymbol{X}_{r(i),j}, i \in [m], j \in [d]$, where $\boldsymbol{X}_{i,j}$ denotes the $j$-th coordinate of $\boldsymbol{X}_i$. Let $\lambda(\cdot)$ denote the eigenvalue of the matrix. Then the eigenvalues of $\boldsymbol{A}^\top \boldsymbol{A}$ can be bounded: with at least $1 - \delta$ probability, $\lambda(\boldsymbol{A}^\top\boldsymbol{A}) \in [1 - \varepsilon_1, 1 + \varepsilon_1]$. (ii) $\sum_{i=1}^{m} \alpha_i/(1 + \varepsilon_2) \leq 1$, and moreover, $\forall i \in [n]$, we have $\alpha_i \max_k \{\sum_{j=1}^{d} \boldsymbol{X}_{k,j}^2/P_{ir(i)}\} \leq \varepsilon_3$.

It is a novel active sampling procedure, blessed by four properties: (i) *Preserving Geometric Structure.* By constraining the eigenvalues of $\boldsymbol{A}^\top\boldsymbol{A}$ near 1, the algorithm ensures that key geometric properties (e.g., the covariance structure) of the original data are retained in the reduced sample. This allows the small subset of points to be a reliable proxy for the entire dataset. (ii) *Selecting Influential Points.* The sampling probabilities $P_{ij}$ typically reflect leverage scores or importance weights, emphasizing the most critical data points for the regression. As a result, even a limited number of samples can capture the most "informative" observations in the feature space. (iii) *Correcting for Sampling Bias.* The reweighting term $w_i = \frac{\alpha_i}{P_{ir(i)}}$ compensates for uneven sampling probabilities. It assigns smaller weights to frequently sampled points and larger weights to rarely sampled points, thereby mitigating bias and controlling variance under a finite-sample regime. (iv) *Enforcing Bounded Influence.* The additional constraints in (ii)—particularly limiting $\sum \alpha_i$ and bounding $\max_k \left\{\sum_{j=1}^{d} \boldsymbol{X}_{k,j}^2/P_{ir(i)}\right\}$—prevent any single point (or small group of points) from dominating the estimation. Combined with the geometric and statistical safeguards above, this bounded influence preserves estimation accuracy despite a significantly reduced sample size. Blessed with properties (i)-(iv), we provide a variant of Theorem 7 of Chen & Price (2019) as follows.

**Lemma 4.2** (approximation error of active sampling). *Given dataset $\{\boldsymbol{X}_i, Y_i\}_{i\in[n]}$ and define $f^{opt} := \arg\min_{h\in\mathcal{H}} \sum_{i=1}^{n} (Y_i - h(\boldsymbol{X}_i))^2$. Moreover, for a good $\varepsilon$-reweighing sampling strategy, $f^{act} := \arg\min_{h\in\mathcal{H}} \sum_{i=1}^{m} w_i(Y_{r(i)} - h(\boldsymbol{X}_{r(i)}))^2$. Then $\forall i \in \{1,\ldots,m\}$, with probability $1 - \delta$,*

$$\sum_{j=1}^{n} P_{ij}(f^{act}(\boldsymbol{X}_j) - f^{opt}(\boldsymbol{X}_j))^2 \leq \frac{\eta}{n}\sum_{j=1}^{n}(f^{opt}(\boldsymbol{X}_j) - Y_i)^2.$$

Here $\eta = (1 + \varepsilon_2)\varepsilon_3 \|\boldsymbol{X}\|_F^2/(1 - \varepsilon_1)$ is a small constant[2] with related to $\varepsilon_1, \varepsilon_2, \varepsilon_3$. We refer readers to Appendix A

---

[2]Here $\|\cdot\|_F$ denotes the Frobenius norm of the matrix.

for the proof details. Lemma 4.2 indicates the approximation error could be controlled via such active sampling process in Algorithm 2. Furthermore, we emphasize that IRD (Algorithm 2) is superior to previous active sampling techniques such as leverage score sampling (Ghadiri et al., 2024; Addanki et al., 2022) in terms of sample complexity. We formulate it in Lemma 4.3.

**Lemma 4.3.** *With probability at least $1 - \delta$, where $\delta$ is a small constant, Algorithm 2 is a good $\varepsilon$-reweighting sampling strategy and is terminated within $O(d/\varepsilon_3)$ iterations.*

The proof is shown in Appendix B. Such sample complexity is also demonstrated to be nearly optimal (up to a constant coefficient) in Section 5.

After this preparation, we propose the total active sampling algorithm in experimental design, which is called ATE estimation via **Re-Weighting Active Sampling** (RWAS estimator). Our inspiration comes from Harshaw et al. (2024); Ghadiri et al. (2024); Addanki et al. (2022). Unlike the objective of Harshaw et al. (2024), we seek to apply their GSW, a well-known estimator with finite-sample guarantees mentioned in our introduction, in a finite-sample setting. We then leverage the coefficients and related covariates learned from our aforementioned IRD algorithm to perform regression adjustment, thereby improving the estimator's accuracy. However, since the IRD algorithm assigns different importance weights to each individual during its iterative process, using a subset of covariates to adjust the individual's own potential outcome is often biased Ghadiri et al. (2024). Therefore, we follow Ghadiri et al. (2024)'s idea by splitting the entire population into two parts, $\{S, \bar{S}\}$, and the total procedure is detailed in Algorithm 1. Here $S$ is used for Bernoulli active sampling to learn the regression coefficients (line **1-4**), while $\bar{S}$ is used to design GSW (line **5**). Based on this partition, we can construct an unbiased estimator (line **6-7**). Note that our main difference from Ghadiri et al. (2024); Addanki et al. (2022) lies in adopting a more efficient strategy (in terms of sample complexity) in $S$ compared to Leveage score sampling, enabling us to achieve the lowest sample complexity so far for a regression-adjusted estimator that induces finite-sample guarantees[3].

### 4.2. Finite-sample bound of ITE & ATE estimation

In this part, we aim to provide the upper bound of ATE under Algorithm 1. For preparation, it is natural first to present a coarse upper bound of ITE estimation.

**Lemma 4.4** (ITE estimation upper bound). *Conditioning that $\forall i, w_i \geq \underline{w} > 0$. Under Algorithm 1, the ITE estimation*

---

[3]In Algorithm 1, we define $\bar{\boldsymbol{X}}_* := \frac{1}{\text{card}(*)}\sum_{i\in*} \boldsymbol{X}_i$, where $* \in [n]$, which is the average of the corresponding row values.

---

**Algorithm 1** RWAS estimator

---

**Require:** $\boldsymbol{X} \in \mathbb{R}^{n*d}, \boldsymbol{Y}^{(1)}, \boldsymbol{Y}^{(0)} \in \mathbb{R}^n.\ \varepsilon, \phi \in (0,1).$

1: $\boldsymbol{Y} = \mathbb{I}(\boldsymbol{Z}=1)\boldsymbol{Y}^{(1)} - \mathbb{I}(\boldsymbol{Z}=0)\boldsymbol{Y}^{(0)}$, where $\mathbb{P}(\boldsymbol{Z}=1) = 0.5.$

2: Let $\tilde{\boldsymbol{X}} = \begin{bmatrix} \boldsymbol{1} & \boldsymbol{X}-\bar{\boldsymbol{X}} \end{bmatrix}$, where $\bar{\boldsymbol{X}}$ is the column-mean matrix of $\boldsymbol{X}$.

3: Select $|S|$ samples via IRD and induces the weights $\{w_1, w_2, ...w_{|S|}\}$.

4: Compute the coefficient in IRD sampling via $\tilde{\boldsymbol{\beta}}^{act} = \arg\min_{\boldsymbol{\beta} \in \mathcal{R}^{d+1}} \|\boldsymbol{SY} - \boldsymbol{S}\tilde{\boldsymbol{X}}\boldsymbol{\beta}\|_2^2$, where $\boldsymbol{S}$ is a diagonal matrix with the $j$-th element $s_j = \sqrt{w_i}$ if there exists $j$ such that $r(i) = j$, and $s_j = 0$ if $j \notin S$. Let $\hat{\boldsymbol{\beta}}^{act}$ be the first d-dimension of $\tilde{\boldsymbol{\beta}}^{act}$ removing the intercept.

5: Let $\hat{\tau}_S = 2\sum_{i \in S} \boldsymbol{Y}(\mathbb{I}(\boldsymbol{Z}=1) - \mathbb{I}(\boldsymbol{Z}=0))/|S|$ be the HT estimator.

6: Uniformly sample $m'$ samples $\bar{S}_{m'} \in \bar{S} = [n]/S.\ |\bar{S}_{m'}| = m'$. Obtain $\hat{\tau}_{\bar{S}}$ by HT estimator and deploy GSW design on $\bar{S}_{m'}$ with parameter $\phi$.

7: Compute $\hat{\tau}_{act,1} = \hat{\tau}_{\bar{S}} - \frac{1}{|\bar{S}_m|} \sum_{i \in \bar{S}_{m'}} \left(\boldsymbol{X}_i - \bar{\boldsymbol{X}}_{\bar{S}_{m'}}\right)^\top \hat{\boldsymbol{\beta}}^{act}$, and let $\hat{\tau}_{act,2} = \hat{\tau}_S$.

**Ensure:** $\hat{\tau}_{act} = (|\bar{S}|\hat{\tau}_{act,1} + |S|\hat{\tau}_{act,2})/n.$

---

| | Sample complexity |
|---|---|
| Addanki et al. (2022) | $\mathcal{O}(d\log(d) + d/\varepsilon)$ |
| Harshaw et al. (2024) | $\mathcal{O}(n)$ |
| Ghadiri et al. (2024) | $\mathcal{O}(d\log(d)/\varepsilon^2 + m')$ |
| **Ours** | $\mathcal{O}(d/\varepsilon + m')$ |

*Table 1.* Comparison of active sampling method in ATE estimation with finite-sample guarantee. Here $\varepsilon$ denotes the relative approximation error, and $m'$ is an independent constant lower than $n$.

bound $\|\tilde{\boldsymbol{X}}\hat{\boldsymbol{\beta}}_{act} - \mathbf{t}\|_2^2$ could be upper bounded by

$$(1+\eta)\min\left\|\tilde{\boldsymbol{X}}\boldsymbol{\beta} - \boldsymbol{t}\right\|_2^2 + \frac{d\|\boldsymbol{\mu}\|_\infty^2}{\underline{w}}.$$

with $(1-\delta)$ probability. Here $w_i$ is a constant.

Here $\eta$ is identified in Lemma 4.2. We defer the proof in Appendix C, where we also justify that the condition in Lemma 4.4 is natural to satisfy. Lemma 4.4 establishes the relationship between the ITE estimation error, fitting precision and the dataset scale. According to it, we further construct the finite-sample ATE bound as follows:

**Theorem 4.5** (ATE estimation upper bound). *Under Algorithm 1, the variance of ATE estimator could be bounded by*

$$\mathbb{E}[(\hat{\tau}_{act} - \tau)^2] \leq \text{error}_{\bar{S}} + \text{error}_{\bar{S},\text{GSW}} + \text{error}_{\bar{S},\text{ITE}}, \text{ where}$$

$$\text{error}_S = \frac{Cd\|\boldsymbol{\mu}\|_\infty^2}{\varepsilon n^2},$$

$$\text{error}_{\bar{S},\text{GSW}} = \min_{\boldsymbol{\beta}}\left[\frac{\left\|\tilde{\boldsymbol{X}}\boldsymbol{\beta} - \boldsymbol{\mu}\right\|_2^2}{m'n\phi} + \frac{1}{(m')^2}\frac{\zeta^2}{1-\phi}\|\boldsymbol{\beta}\|_2^2\right],$$

$$\text{error}_{\bar{S},\text{ITE}} = \frac{1+\eta}{m'n}\min_{\boldsymbol{\beta}}\left\|\tilde{\boldsymbol{X}}\boldsymbol{\beta} - (\boldsymbol{t}-\bar{\boldsymbol{t}})\right\|_2^2 + \frac{d\|\boldsymbol{\mu}\|_\infty^2}{m'n\underline{w}}.$$

with $(1-\delta)$ probability under Algorithm 1.

---

**Algorithm 2** IRD, modified from Chen & Price (2019)

---

**Require:** Data $\boldsymbol{X} = (\boldsymbol{X}_1, \ldots, \boldsymbol{X}_n)^\top.$

1: Initialize: $\gamma = \min\{\sqrt{\varepsilon}/\sqrt{3.41}, 1\}, i = 0, B_0 = \boldsymbol{0}, l_0 = -2d/\gamma, u_0 = 2d/\gamma.$

2: **while** $u_i - l_i < 8d/\gamma$ **do**

3:     Identify the coefficients $\Phi_i = tr(u_iI - B_i)^{-1} + tr(B_i - l_iI)^{-1}, \alpha_i = (\gamma/\Phi_i)*(\gamma^2/2d(1-\gamma^2));$

4:     $P_{ij} = \boldsymbol{X}_j^\top \left[(u_iI - B_i)^{-1} + (B_i - l_iI)^{-1}\right] \boldsymbol{X}_j/\Phi_i;$

5:     Sample $r(i) = j$ with probability $P_{ij}$, and let $\boldsymbol{X}_{r(i)} = \boldsymbol{X}_j;$

6:     Update the matrix $B_{i+1} = B_i + \gamma/(\Phi_i P_{ir(i)}) * \boldsymbol{X}_{r(i)}\boldsymbol{X}_{r(i)}^\top;$

7:     Update the upper and lower bounds $u_{i+1} = u_i + \Phi_i^{-1}\gamma/(1-\gamma), l_{i+1} = l_i + \Phi_i^{-1}\gamma/(1+\gamma), i = i+1;$

8: **end while**

**Ensure:** Terminates at $i = m$, and produces $\{w_i = \alpha_i/P_{ir(i)}\}_{i \in [m]}$ for each iteration.

---

Theorem 4.5 reveals that the ATE finite-sample estimation error inherently consists of three parts, deriving from two partitions: for $S$, it induces the upper bound according to the naive HT estimator; on the other hand, for $\bar{S}$, it causes another two kinds of upper bound, corresponding to the GSW design guarantee and Lemma 4.4, respectively.

Recall that a fundamental result is that when we select $n'$ samples from the whole population ($n$ samples), the upper bound of ATE via the traditional Horvitz-Thompson (HT) estimator without covariates is $\mathbb{E}(\hat{\tau} - \tau)^2 \leq \frac{\|\boldsymbol{\mu}\|_2^2}{n'n} + \frac{\|\boldsymbol{t}-\bar{\boldsymbol{t}}\|_2^2}{n'n}$ (Ghadiri et al., 2024; Harshaw et al., 2024), where $\bar{\boldsymbol{T}}$ is the expectation of $\boldsymbol{T}$. For comparison, the first term is optimized via regression-based design technique and formalized as $\text{error}_{S,\text{GSW}}$ and $\text{error}_{S,\text{ITE}}$ using $m'$ samples, and the second term is inherited by $\text{error}_S$, with an diminishing scale of individual set from $n$ to $\mathcal{O}(d/\varepsilon_3)$. We defer a more thorough comparison with bounds in previous literature in

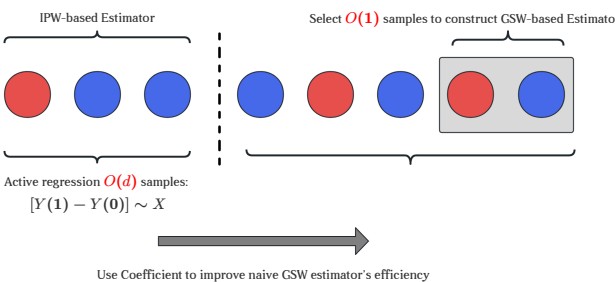

*Figure 1.* Illustration of our main algorithm.

Appendix I.

## 5. Additional discussion: lower bound

In the above section, we propose the RWAS estimator and then analyze its precision. Hence, a natural question arises: what is the lower bound of the query complexity based on the dimension $d$? This section establishes a lower-bound conditioning that constantly bounds the estimation error, under a super-population perspective.

**Theorem 5.1** (Lower bound). *∀ any fixed dimension $d$, for any $\epsilon \in (0, 0.1)$ sufficiently small, there exists a feasible set of $\{\mathbf{Y}^1, \mathbf{Y}^0\}$ such that for any algorithm whose output $\hat{\tau}$ satisfying $|\hat{\tau} - \tau| \le 0.1$ with probability $0.75$, need at least $2.86d/\varepsilon$ sample queries.*

The detailed explanation is deferred into Appendix E. It demonstrates that for active sampling algorithms in experimental design, achieving a satisfactory precision level requires a sample complexity at the linear complexity level, which is not optimizable. This lower bound and the upper bound differ only by a constant factor, making it nearly optimal.

*Remark* 5.2. Despite the seemingly counterintuitive result that estimation errors can be controlled within $0.1$ with some probability while still requiring a sample size of $O(d/\epsilon)$, this outcome is mathematically grounded. Specifically, even under mild estimation error conditions, there exist instances where the estimation process becomes particularly challenging, requiring significantly more samples. We illustrate this with a construction where the individual treatment effect (ITE) $\tau(X_i)$ is defined as a function $L(X_i)$ plus Gaussian noise $\mu$. This setup, while meeting the error bound, demands a sample size of at least $O(d/\epsilon)$. The intuition behind this lies in the existence of a large number of function mappings with a nontrivial distance between them, which complicates the estimation process. As detailed in our proof, this condition, alongside the mutual information bound derived using Fano's inequality and the Shannon-Hartley theorem, results in the necessity of a sample size proportional to the dimensionality $d$ and inversely proportional to the error tolerance $\epsilon$. Thus, our analysis confirms that the sample complexity required for accurate estimation is optimal and cannot be reduced further without sacrificing precision.

## 6. Additional discussion: SUTVA assumption

In the preceding chapters, we introduced an active sampling algorithm with a sample complexity of $\Omega(d)$ (up to a constant factor) and established its near-optimality by deriving a corresponding lower bound for the problem. However, these results rely fundamentally on Assumption 3.1, which asserts that the treatment assigned to one node does not influence the outcomes of other nodes. It is well recognized, however, that this assumption may be violated in many social network settings. For instance, as discussed in the introduction with the COVID-19 vaccine case, an individual's health outcome can be directly affected by the treatment status (e.g., vaccination) of those in their immediate surroundings.

This section extends our previous findings to the network setting by relaxing the SUTVA assumption. Specifically, we develop an active sampling algorithm that achieves a finite-sample error bound when treatments and outcomes exhibit network-based dependencies. This advancement lays the groundwork for analyzing and designing experimental strategies in social network contexts, where interference among interconnected units must be considered.

The concept in the above paragraph is inherited. Moreover, the additional notation is the arbitrary neighborhood interference exposure mapping:

**Definition 6.1.** (Arbitrary neighborhood interference (Kandiros et al., 2024)) We define the arbitrary neighborhood exposure mapping $d_i$ as $d_i(\mathbf{T}) := \{j \in N(i) \cup \{i\} : t_j = 1\}$, representing the index set of treated neighborhoods. Here, $N(i)$ denotes the set of nodes that share an edge with node $i$. Additionally, we denote the adjacency matrix as $\mathbb{H}$.

**Assumption 6.2.** (Additive Potential Outcome (Kandiros et al., 2024)) For every unit $i \in [n]$ and interventions $\mathbf{T}', \mathbf{T}''$, if $d_i(\mathbf{T}') = d_i(\mathbf{T}'')$, then $y_i(\mathbf{T}') = y_i(\mathbf{T}'')$. Equivalently, for each unit $i \in [n]$, there exist coefficients $\alpha_{i,S} \in \mathbb{R}$ for $S \subseteq \widetilde{\mathcal{N}}(i)$ such that the potential outcome function $y_i$ can be expressed as

$$Y_i(\mathbf{T}) = \sum_{S \subseteq \widetilde{\mathcal{N}}(i)} \alpha_{i,S} \cdot \prod_{j \in S} t_j \prod_{j \notin S} (1 - t_j).$$

In this sense, our causal estimands are defined as

$$\tau_i = Y_i(\mathbf{Z}_{i,(1)}) - Y_i(\mathbf{Z}_{i,(0)}), \quad \tau = \frac{1}{n} \sum_{i \in [n]} \tau_i.$$

Analogous to Kandiros et al. (2024), we could artificially set the definition of exposure mapping $d_i(\mathbf{Z})$ to degenerate to

the well-known concepts such as global ATE, direct effect and spill-over effect. Following the conflict graph construction strategy in Kandiros et al. (2024), the conflict graph design is in Algorithm 4, which is deferred in Appendix H. Under such design, the current estimator is represented as

$$\hat{\tau} = \frac{1}{n} \sum_{i=1}^{n} (Y_i - \boldsymbol{X}_i^{\mathbf{G}} \boldsymbol{\beta}) \left( \frac{\mathbf{1}\left[E_{(i,1)}\right]}{P\left(E_{(i,1)}\right)} - \frac{\mathbf{1}\left[E_{(i,0)}\right]}{P\left(E_{(i,0)}\right)} \right).$$
(1)

Here $E_{(i,k)} = \{U_i = e_k$ and $U_j = *$ for all $j \in \mathcal{N}_b^{\pi}(i)\}$ according to Kandiros et al. (2024).

Then, we provide the thorough active sampling algorithm under such network interference network, which is called ATE estimation via **C**onflict-**G**raph **A**ctive **S**ampling (CGAS) in Algorithm 3. Here $X_i^{\mathbf{G}} := \mathscr{A}(X_i, \mathbb{H})$ is the aggregated function, which could be realized by graph neural network such as Leung (2022); Ma & Tresp (2021); Ma et al. (2022).

We provide the result as follows:

**Proposition 6.3.** *Under Algorithm 6, the variance of estimator could be bounded by* $\mathbb{E}[(\hat{\tau}_{act} - \tau)^2] \leq \frac{2d^2}{\varepsilon^2 n^2} C + \frac{25\lambda(\mathcal{H})}{n^2} (\frac{1}{2}\|\boldsymbol{\mu}\|^2 + (1 + \eta) \min \left\|\boldsymbol{X}^{\boldsymbol{G}} \boldsymbol{\beta} - \frac{1}{2}\boldsymbol{t}\right\|_2^2 + \frac{md^2\|\boldsymbol{\mu}\|_\infty^2 \|\boldsymbol{X}\|_\infty^2}{1 - \varepsilon_1})^2,$ *where $C$ is a constant.*

We defer the proof into Appendix F. Proposition 6.3 indicates that our active sampling technique is an intuitive and generalizable method that seamlessly integrates with network settings. Specifically, it allows for direct regression adjustment on the latest finite-sample results to enhance estimation accuracy. In contrast to the conclusions in Section 4, we do not claim that our results are guaranteed unbiased. This, in turn, reinforces the key insight from Kandiros et al. (2024): in the absence of additional structural assumptions about the network, estimators inherently face a bias-variance trade-off—optimizing for minimal variance alone may introduce significant bias. Striking an optimal balance between these two competing factors remains an important direction for our future work.

# 7. Experiment

In this section, we conduct synthetic and real-world experiments to demonstrate the unbiasedness and efficiency of our estimator. We aim to present our **RWAS** estimator could outperform the competitive baseline in terms of unbiasedness and precision, Whether these methods achieve a theoretical finite-sample error bound (e.g., Ghadiri et al. (2024); Addanki et al. (2022)) or only possess asymptotic normality (e.g., HT estimator).

Most of the baselines are inherited from Ghadiri et al. (2024). The difference is that we focus on ATE estimation based on limited, finite samples instead of the whole population. Specifically, we compare the following baselines in our ex-

periments: (i) **HT** estimator, which employs the traditional Horvitz-Thompson estimator (Horvitz & Thompson, 1952) taking advantage of the inverse propensity score. (ii) **Hajek** estimator, another commonly-used estimator in ATE estimation attributed to Hajek (1971). (iii) **Classic Regression Adjustment (CRA)** estimator, which adopts the method in Li & Ding (2020). In (i)-(iii), the estimators are operated under finite samples $m$ among the total population via the random Bernoulli trial with probability 0.5. (iv) **Gram-Schimit Walk design (GSW)** The HT estimator following the GSW design in Harshaw et al. (2024)[4]. (v) **Regression Adjusted Horvitz-Thompson (RAHT)** estimator proposed in Ghadiri et al. (2024), a method combining GSW design and leverage score sampling. (vi) **Sample-constrained (SC)** estimator in Addanki et al. (2022). Also, **4-Vectors** by Mou et al. (2022), which also adopts similar cross-adjustment strategy. To ensure fairness, we keep the same amount of selected samples in all these settings except GSW-Adj. For GSW-Adj, the sample size is selected as the maximal number of samples allocated for GSW design among all other methods. To evaluate the performances of methods in finite samples, we repeat the method holding the sample fixed, which is generated in the initial of the experiment.

## 7.1. Synthetic Experiment

To estimate ATE, the procedure and data-generating process of the experiment are illustrated as follows:

**ATE Dataset** Denote the sample size as $n$ and set the amount of covariates $d = 50$. The matrix of covariates $\mathbf{X} \in \mathbb{R}^{n \times d}$ is generated in three steps. First, generate matrix $\tilde{X} \in \mathbb{R}^{n \times d}$, with each entry sampled from uniform distribution in $[0, 0.01]$ independently. Then, a Gram-Schmidt orthogonalization process is performed on the column space of $\tilde{X}$ to generate an orthogonal matrix $\mathbf{Q} \in \mathbb{R}^{n \times d}$ satisfying $\mathbf{Q}^\top \mathbf{Q} = \mathbf{I}_d$. Finally, set $\mathbf{X} = n/10 * \mathbf{Q}$ to recover the column norm. The potential outcome vector for the control $\mathbf{y}^{(0)}$ is generated uniformly at random from $[0, 5]$, and the individual treatment effect vector $\boldsymbol{t}$ satisfies $\boldsymbol{t} = \mathbf{X}\mathbf{b} + \mathbf{r}$, with each element of $\mathbf{b} \in \mathbb{R}^d$ be a uniform random number in $[0, 1]$, and $\mathbf{r} \in \mathbb{R}^n$ follows a mean zero Gaussian distribution with a standard deviation $sd = 0.2$. Eventually, $\mathbf{y}^{(1)}$ is generated by $\boldsymbol{t} = \mathbf{y}^{(1)} - \mathbf{y}^{(0)}$. The ground truth of ATE is set as $\tau = \frac{1}{n} \sum_{i \in [n]} t_i$, with $\boldsymbol{t} = (t_1, \ldots, t_n)$.

**Experiment Procedure** Generate an ATE dataset with size $n$ at the start to serve as pre-determined finite samples. Then, run each relevant method $r$ rounds and get the esti-

---

[4]We also conduct naive regression adjustment upon it if its performance could be better, for better competiveness of such baseline.

---

**Algorithm 3** CGAS

---

**Require:** $X \in \mathbb{R}^{n*d}$, $\mathbb{H}$, $\boldsymbol{Y}^{(1)}, \boldsymbol{Y}^{(0)} \in \mathbb{R}^n$. $\varepsilon, \phi \in (0,1)$.

1: $\boldsymbol{Y} = \mathbb{I}(E_{(i,1)} = 1)\boldsymbol{Y}^{(1)} - \mathbb{I}(E_{(i,0)} = 0)\boldsymbol{Y}^{(0)}$, where $\mathbb{P}(\boldsymbol{Z} = 1) = 0.5$.

2: Select $|S|$ samples via IRD $\{\frac{1}{2}\boldsymbol{Y}, \boldsymbol{X}^{\mathbf{G}}\}$ and induces the weights $\{w_1, w_2, ...w_{|S|}\}$.

3: Compute the coefficient in active reweighing sampling via $\tilde{\boldsymbol{\beta}}^{act} = argmin_{\boldsymbol{\beta} \in \mathcal{R}^{d+1}} \|\boldsymbol{S}\boldsymbol{Y} - \boldsymbol{S}\boldsymbol{X}\boldsymbol{\beta}\|$, where $\boldsymbol{S}$ is a diagonal matrix with the $j$-th element $s_j = \sqrt{w_i}$ if there exists $j$ such that $r(i) = j$, and $s_j = 0$ if $j \notin S$.

4: Set an estimator $\hat{\tau}_S$ to estimate $\sum_{i \in S} Y_i(1) - Y_i(0)$ (could be manually adjusted).

5: Uniformly sample $m'$ samples $\bar{S}_{m'} \in \bar{S} = [n]/S$. $|\bar{S}_{m'}| = m'$.For the complement set, conduct **CGD** in Algorithm 4 and employ the assignments in to the estimator in (1). Namely, $\hat{\tau}_{\bar{S}} := \frac{1}{|\bar{S}_{m'}|}\sum_{i \in \bar{S}_{m'}}(Y_i - \boldsymbol{X}_i^{\mathbf{G}}\boldsymbol{\beta}^{act})\left(\frac{\mathbf{1}[E_{(i,1)}]}{P(E_{(i,1)})} - \frac{\mathbf{1}[E_{(i,0)}]}{P(E_{(i,0)})}\right)$.

6: Replicate Line $6 - 7$ in Algorithm 1.

**Ensure:** $\hat{\tau}_{act} = (|\bar{S}|\hat{\tau}_{act,1} + |S|\hat{\tau}_{act,2})/n$.

---

*Table 2.* **Upper:** Synthetic experiment: Error(sd) of ATE estimations. Prop. = Proportion of samples used in estimation. sd = Standard deviation of estimation. The bolded number denotes the smallest absolute bias. The reported errors are multiplied by 10. **Lower:** Real-world experiments: Error(sd) of ATE estimations. For the final line, we report the result with scale $e^{-3}$.

| Sample size | Prop. | HT | Hajek | CRA | GSW | RAHT | SC | 4-Vectors | RWAS (Ours) |
|---|---|---|---|---|---|---|---|---|---|
| | 0.2 | 1.99 (1.24) | 1.89 (1.09) | 2.62 (0.66) | 2.39 (1.17) | **1.81 (1.19)** | 2.54 (1.54) | 1.99 (1.43) | 1.84 (1.16) |
| 1000 | 0.5 | 1.14 (0.74) | 1.03 (0.50) | 1.02 (0.39) | 0.80 (0.64) | 0.80 (0.56) | 0.80 (0.68) | 0.92 (0.61) | **0.71 (0.47)** |
| | 0.8 | 0.86 (0.54) | 0.69 (0.24) | 0.64 (0.26) | **0.40 (0.44)** | 0.43 (0.28) | 0.49 (0.52) | 0.51 (0.33) | **0.40 (0.26)** |
| | 0.2 | 2.01 (1.41) | 1.76 (0.99) | 1.58 (0.74) | 1.54 (1.24) | 1.91 (1.30) | 1.76 (1.43) | 1.43 (0.97) | **1.29 (0.88)** |
| 2000 | 0.5 | 0.95 (0.66) | 0.91 (0.40) | 0.70 (0.35) | **0.58 (0.62)** | 0.70 (0.49) | 0.71 (0.42) | 0.75 (0.49) | 0.59 (0.41) |
| | 0.8 | 0.66 (0.45) | 0.52 (0.21) | 0.43 (0.22) | **0.30 (0.37)** | 0.32 (0.23) | 0.38 (0.35) | 0.39 (0.42) | **0.30 (0.22)** |

| Dataset | Prop. | HT | Hajek | CRA | GSW | RAHT | SC | 4-Vectors | RWAS (Ours) |
|---|---|---|---|---|---|---|---|---|---|
| | 0.2 | 2.35 (1.92) | 2.40 (2.10) | **0.80 (0.72)** | 1.15 (1.00) | 1.05 (0.62) | 1.08 (0.96) | 2.30 (1.94) | 0.99 (0.80) |
| Boston | 0.5 | 1.98 (1.63) | 2.02 (1.81) | 0.68 (0.56) | 0.95 (0.84) | 0.89 (0.50) | 0.91 (0.80) | 1.95 (1.65) | **0.65 (1.55)** |
| | 0.8 | 1.79 (1.45) | 1.80 (1.65) | **0.54 (0.43)** | 0.78 (0.69) | 0.75 (0.41) | 0.75 (0.65) | 1.76 (1.54) | 0.74 (0.43) |
| | 0.2 | 0.42 (0.38) | 0.40 (0.35) | 0.06 (0.04) | 0.08 (0.06) | 0.08 (0.06) | 0.95 (0.23) | 0.44 (0.33) | **0.07 (0.06)** |
| IHDP | 0.5 | 0.35 (0.31) | 0.32 (0.29) | 0.04 (0.03) | 0.07 (0.05) | **0.06 (0.05)** | 0.75 (0.18) | 0.36 (0.27) | **0.06 (0.05)** |
| | 0.8 | 0.29 (0.26) | 0.26 (0.25) | 0.02 (0.01) | 0.05 (0.03) | **0.05 (0.04)** | 0.62 (0.12) | 0.30 (0.21) | **0.05 (0.04)** |
| | 0.2 | 2.10 (1.68) | 1.95 (1.50) | **1.45 (1.09)** | 1.45 (1.35) | 1.61 (1.23) | 1.92 (1.58) | 2.07 (1.74) | 1.65 (1.54) |
| Lalonde | 0.5 | 1.82 (1.48) | 1.73 (1.34) | 1.15 (0.87) | 1.36 (1.09) | 1.30 (1.00) | 1.57 (1.32) | 1.80 (1.50) | **1.13 (0.88)** |
| | 0.8 | 1.60 (1.22) | 1.48 (1.16) | **0.94 (0.70)** | 1.08 (0.87) | 1.05 (0.79) | 1.32 (1.05) | 1.57 (1.25) | 1.04 (0.70) |
| | 0.2 | 1.90 (1.42) | 2.00 (1.37) | **1.55 (1.23)** | 1.80 (1.43) | 1.70 (1.38) | 1.88 (1.50) | 1.85 (1.53) | **1.55 (1.43)** |
| Twins | 0.5 | 1.65 (1.24) | 1.73 (1.18) | 1.35 (1.08) | 1.55 (1.27) | 1.48 (1.22) | 1.61 (1.35) | 1.60 (1.38) | **1.21 (1.32)** |
| | 0.8 | 1.42 (1.04) | 1.51 (0.97) | **1.21 (0.99)** | 1.34 (1.03) | 1.24 (0.99) | 1.39 (1.10) | 1.40 (1.12) | **1.21 (1.04)** |

mate $\hat{\tau}_j$ for ATE $\tau$ in round $j$. The performance is estimated through Error $= \sum_{j \in [r]} |\hat{\tau}_j - \tau|/r|\tau|$.

**Result Analysis** Table 7 illustrates the performance of methods with different sample sizes and proportions of selected samples. Compared with classical estimators under the Bernoulli trial, our estimator generally has lower estimation error, with an increasing gap as the proportion of selected samples enlarges. Our method has an inherited trade-off between covariate balance and robustness. In finite samples, robustness is reached once sufficient samples are selected in the experiment. Therefore, our method achieves more balanced covariates among treatment and control groups compared with a completely randomized assignment, which benefits the performance. Among the estimators with balancing assignments (**GSW-Adj,RAHT,RWAS**), our

*Table 3.* Characteristics of different real-world datasets. To enable evaluation, we set a true homogeneous ATE for each dataset based on relevant results. No. Feature: Dimension of features (covariates). Semi-synthetic means to be computed based on manually pre-defined fixed shift.

| | Boston | IHDP | Twins | LaLonde |
|---|---|---|---|---|
| Sample Size | 506 | 747 | 32,120 | 445 |
| No. Feature | 13 | 25 | 50 | 10 |
| True ATE | 0 | -4.016 | $6.4 \times e^{-3}$ | Semi-synthetic |

estimator generally has lower error compared with **RAHT**, which indicates our method is pretty competitive among relevant methods.

## 7.2. Real-world Experiment

We evaluate the performances of methods on the following real-world datasets: Boston Dataset Harrison Jr & Rubinfeld (1978), IHDP Dataset Multisite (1990); Dorie (2016), Twins Dataset Almond et al. (2005) and LaLonde Dataset LaLonde (1986), whose basic information are shown in Table 7.1.

It illustrated that our method outperforms the previous baselines in most cases: (i) **Exponential Decay in Estimation Error and Variance with Increasing Active Samples**. As the number of actively sampled instances increases, both the estimation error and variance exhibit an exponential decay trend. In particular, when the sample size is relatively small (e.g., only 20% proportion), the approximation error in causal queries is susceptible to the number of samples. This further reinforces our motivation to seek the optimal sample complexity in active sampling algorithms—since sample size significantly impacts model performance. (ii) **Comparable Performance of CRA and RWAS**, but with Theoretical Limitations. In many scenarios, **CRA** performs comparably to our RWAS, suggesting it can be highly effective in practice. However, such strong empirical performance remains largely anecdotal, as **CRA** lacks a well-established finite-sample theoretical analysis. This highlights a key limitation of **CRA** when contrasted with theoretically grounded active sampling techniques. (iii) **GSW's Performance and Its Trade-offs**. Additionally, the performance of **GSW** is worth noting. As acknowledged by Ghadiri et al. (2024), the GSW approach often surpasses **RAHT**—and, at times (such as in the Lalonde and Twins dataset), even outperforms our **RWAS**. However, **RAHT** and **RWAS** remain of clear research significance because **GSW** typically requires a substantially higher computational complexity. Moreover, since **GSW** relies on offline balancing of covariates across the entire population, it is inherently less adaptable to online settings. In contrast, both **RAHT** and **RWAS** are more readily extendable to online applications, making them more versatile in dynamic experimental designs.

## 8. Conclusion and Discussion

Our paper establishes a new finite-sample framework for active sampling in causal inference. It advances experimental design's theoretical and practical understanding, opening new directions for efficient design in constrained settings. For future work, it would be promising to explore further these two trade-offs: (i) the accuracy-robustness trade-off, which would inspire more advancing estimators beyond `GSW` design, especially for the regression-adjustment cases; (ii) the bias-variance trade-off, it would shed on more insights on developing estimators in the network interference.

## Acknowledgements

This work is supported by National Key R&D Program of China (2022ZD0160300) and the NSF China (No. 62276004, 623B2002).

## Impact Statement

This paper presents work whose goal is to advance the field of Machine Learning. There are many potential societal consequences of our work, none which we feel must be specifically highlighted here.

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

# Supplementary Material for "Active Treatment Effect Estimation via Limited Samples"

For the sake of simplicity and without introducing any misleading implications, we denote the additional sampled quantity $m'$ in GSW as $m$ in the main text.

## A. The proof of lemma. 4.2

**Proof.** We follow the notation and analogy in Chen & Price (2019). Notice that

$$f^{act} := \arg\min_{h \in \mathcal{F}} \frac{1}{m} \sum_{i=1}^{m} w_i (Y_i - h(\boldsymbol{X}_{r(i)}))^2. \tag{2}$$

We consider the coefficient of function $f^{act}$. If we use a $d * 1$ vector $\alpha(\cdot)$ to denote the coefficient of function on each orthonormal basis, then we have

$$\alpha(f^{act}) = \arg\min_{\alpha(h)} \|A * \alpha(h) - \boldsymbol{w} \circ \boldsymbol{Y_m}\|_2 = \|(A^\top A)^{-1} A^\top (\boldsymbol{w} \circ \boldsymbol{Y_m})\|_2.$$

$$\alpha(f^{opt}) = \|(A^\top A)^{-1} A^\top (\boldsymbol{w} \circ \boldsymbol{f_m})\|_2. \tag{3}$$

Here $A$ is identified in our main text, $\boldsymbol{w} = (\sqrt{w_1}, \sqrt{w_2}, ... \sqrt{w_m})^\top$, $\boldsymbol{Y_m} = (Y_{r(1)}, Y_{r(2)}, ... Y_{r(m)})^\top$, $\boldsymbol{f_m} = (f^{opt}(\boldsymbol{X}_{r_{(1)}}), ... f^{opt}(\boldsymbol{X}_{r_{(m)}}))^\top$, and $\circ$ denotes the Hadamard product. Then we have

$$\frac{1}{n} \sum_{i=1}^{n} \left(f^{act}(\boldsymbol{X}_i) - f^{opt}(\boldsymbol{X}_i)\right)^2 = \frac{1}{n} \|\boldsymbol{X}[\alpha(f^{act}) - \alpha(f^{opt})]\|_2^2 \le \frac{1}{n} \|\boldsymbol{X}\|_2^2 \cdot \|\alpha(f^{act}) - \alpha(f^{opt})\|_2^2. \tag{4}$$

According to Eq. (3)-(4),

$$\begin{aligned}
\mathbb{E}_r \|\alpha(f^{act}) - \alpha(f^{opt})\|_2^2 &= \mathbb{E}_r \|(A^\top A)^{-1} A^\top \boldsymbol{w} \circ (\boldsymbol{Y_m} - \boldsymbol{f_m})\|_2^2 \\
&\le \mathbb{E}_r \|(A^\top A)^{-1}\|_2^2 * \mathbb{E}_r \|A^\top \boldsymbol{w} \circ (\boldsymbol{Y_m} - \boldsymbol{f_m})\|_2^2 \\
&\le \mathbb{E}_r \|(A^\top A)^{-1}\|_2^2 * \mathbb{E}_r \sum_{j=1}^{d} \left( \sum_{i=1}^{m} w_i \boldsymbol{X}_{r(i),j} (f^{opt}(\boldsymbol{X}_{r(i)}) - Y_{r(i)}) \right)^2 \\
&= \mathbb{E}_r \|(A^\top A)^{-1}\|_2^2 * \mathbb{E}_r \sum_{j=1}^{d} \sum_{i=1}^{m} w_i^2 \boldsymbol{X}_{r(i),j}^2 (f^{opt}(\boldsymbol{X}_{r(i)}) - Y_{r(i)})^2,
\end{aligned} \tag{5}$$

where $\mathbb{E}_r$ denotes the expectation taken upon sampling. Since $(1, 1, \ldots, 1)_n$ is included in the column space of $\boldsymbol{X}$, the last equality comes from the fact that

$$\mathbb{E}_r w_i \boldsymbol{X}_{r(i),j} (f^{opt}(\boldsymbol{X}_{r(i)}) - Y_{r(i)}) = \sum_{k=1}^{n} \frac{\alpha_i P_{ik}}{P_{ik}} \boldsymbol{X}_{k,j} (f^{opt}(\boldsymbol{X}_k) - Y_k) = 0. \tag{6}$$

Hence

$$\begin{aligned}
&\mathbb{E}_r \|\alpha(f^{act}) - \alpha(f^{opt})\|_2^2 \\
\le& \mathbb{E}_r \|(A^\top A)^{-1}\|_2^2 * \mathbb{E}_r \left( \max_{i \in [m], k \in [n]} w_i \sum_{j=1}^{d} \boldsymbol{X}_{k,j}^2 \right) \left( \sum_{i=1}^{m} \frac{\alpha_i}{P_{ir(i)}} (f^{opt}(\boldsymbol{X}_{r(i)}) - Y_{r(i)})^2 \right) \\
=& \mathbb{E}_r \|(A^\top A)^{-1}\|_2^2 * \mathbb{E}_r \left( \max_{i \in [m], k \in [n]} w_i \sum_{j=1}^{d} \boldsymbol{X}_{k,j}^2 \right) \left( \sum_{i=1}^{m} \sum_{k=1}^{n} \alpha_i (f^{opt}(\boldsymbol{X}_k) - Y_k)^2 \right) \\
=& \mathbb{E}_r \left( \|(A^\top A)^{-1}\|_2^2 \sum_{i=1}^{m} \alpha_i \right) * \mathbb{E}_r \left( \max_{i \in [m], k \in [n]} w_i \sum_{j=1}^{d} \boldsymbol{X}_{k,j}^2 \right) \left( \sum_{k=1}^{n} (f^{opt}(\boldsymbol{X}_k) - Y_k)^2 \right).
\end{aligned} \tag{7}$$

Observe that $\mathbb{E}_r \|(A^\top A)^{-1}\|_2^2 \leq \frac{1}{1-\varepsilon_1}$ on event $\lambda(A^\top A) \in (1-\varepsilon_1, 1+\varepsilon_1)$, and

$$\max_{i \in [m], k \in [n]} w_i \sum_{j=1}^{d} \boldsymbol{X}_{k,j}^2 = \max_{i \in [m], k \in [n]} \sum_{j=1}^{d} \frac{\alpha_i}{P_{ir(i)}} \boldsymbol{X}_{k,j}^2 \leq \epsilon_3. \tag{8}$$

Therefore, recalling Eq. (4), we finally get

$$\mathbb{E}_r \| f^{act} - f^{opt} \|^2 \leq \frac{(1+\varepsilon_2)\varepsilon_3 \|\boldsymbol{X}\|_F^2}{n(1-\varepsilon_1)} \sum_{k=1}^{n} (f^{opt}(\boldsymbol{X}_k) - Y_k)^2 \tag{9}$$

on an event with probability at least $1 - \delta$, which proves Lemma 4.2.

## B. The proof of lemma 4.3

We prove Algorithm. 2 is a good $\varepsilon$-reweighting sampling strategy with high probability based on Chen & Price (2019).

**Proof.** It is equivalent to prove that Algorithm. 2 contains two main properties with high probability in our main text.

**For the first property**, notice the construction $A(i,j) = \sqrt{w_i} * \boldsymbol{X}_{r(i),j}, i \in [m], j \in [d]$ of matrix $A$, we have

$$A^\top A = \sum_{i=1}^{m} w_i \boldsymbol{X}_{r(i)} \boldsymbol{X}_{r(i)}^\top = \sum_{i=1}^{m} \frac{\alpha_i}{P_{ir(i)}} \boldsymbol{X}_{r(i)} \boldsymbol{X}_{r(i)}^\top = B_m \frac{\gamma^2}{2d(1-\gamma^2)}. \tag{10}$$

Here $\boldsymbol{X}_{r(i)} = (\boldsymbol{X}_{r(1),1}, \ldots, \boldsymbol{X}_{r(1),d})^\top$, and $B_m$ is defined in Algorithm 2.

According to lemma. G.3, we have

$$\lambda(A^\top A) \in \left( \frac{\gamma^2}{2d(1-\gamma^2)} l_m, \frac{\gamma^2}{2d(1-\gamma^2)} u_m \right). \tag{11}$$

Moreover, due to Algorithm 2, we have

$$\frac{u_m + l_m}{\frac{1}{1-\gamma} + \frac{1}{1+\gamma}} = \sum_{i=1}^{m} \frac{\gamma}{\Phi_i}. \tag{12}$$

On the observations (11), (12) above, in order to substitute $u_m, l_m$ in the equations, we aim to bridge the relationship between $\sum_{i=1}^{m} \frac{\gamma}{\Phi_i}$ and the coefficient $\frac{\gamma^2}{2d(1-\gamma^2)}$ as follows.

Since the **while** loop in Algorithm 2 stops at the $m$-th iteration, we have

$$u_m - l_m = \sum_{i=1}^{m} \frac{\gamma}{\Phi_i} \left( \frac{1}{1-\gamma} - \frac{1}{1+\gamma} \right) + \frac{4d}{\gamma} \geq \frac{8d}{\gamma}. \text{ Hence } \sum_{i=1}^{m} \frac{\gamma}{\Phi_i} \geq \frac{2d(1-\gamma^2)}{\gamma^2}. \tag{13}$$

In the $(m-1)$-th iteration, from the condition of the **while** loop, we have

$$\sum_{i=1}^{m-1} \frac{\gamma}{\Phi_i} \leq \frac{4d}{\gamma} / \left( \frac{1}{1-\gamma} - \frac{1}{1+\gamma} \right) = \frac{2d(1-\gamma^2)}{\gamma^2}. \tag{14}$$

Moreover, since the trace of a matrix equals to the sum of its eigenvalues, we have

$$\Phi_m = \sum_{i=1}^{d} \left[ \frac{1}{u_m - \lambda_i(B_m)} + \frac{1}{\lambda_i(B_m) - l_m} \right] \geq \frac{1}{\sum_{i=1}^{m} (u_m - l_m)} (2d)^2 \geq \frac{1}{\frac{4d}{\gamma}} 4d^2 = d\gamma \tag{15}$$

Therefore

$$\frac{2d(1-\gamma^2)}{\gamma^2} \geq \sum_{i=1}^{m} \frac{\gamma}{\Phi_i} - \frac{\gamma}{\Phi_m} \geq \sum_{i=1}^{m} \frac{\gamma}{\Phi_i} - \frac{1}{d} \implies \sum_{i=1}^{m} \frac{\gamma}{\Phi_i} \leq \frac{2d(1-\gamma^2)}{\gamma^2} + \frac{1}{d}. \tag{16}$$

Combined with Eqn. (12), (14) and (16), we have

$$\frac{2d(1-\gamma^2)}{\gamma^2} \in \left[ \sum_{i=1}^{m} \frac{\gamma}{\Phi_i} - \frac{1}{d}, \sum_{i=1}^{m} \frac{\gamma}{\Phi_i} \right] = \left[ \frac{u_m + l_m}{\frac{1}{1-\gamma} + \frac{1}{1+\gamma}} - \frac{1}{d}, \frac{u_m + l_m}{\frac{1}{1-\gamma} + \frac{1}{1+\gamma}} \right]. \tag{17}$$

Combined with Eqn. (11), we have

$$\lambda(A^\top A) \in \left[ \frac{l_m}{\left( \frac{u_m + l_m}{\frac{1}{1-\gamma} + \frac{1}{1+\gamma}} \right)}, \frac{u_m}{\left( \frac{u_m + l_m}{\frac{1}{1-\gamma} + \frac{1}{1+\gamma}} \right) - \frac{1}{d}} \right] = \left[ \frac{\frac{1}{1-\gamma} + \frac{1}{1+\gamma}}{1 + \frac{u_m}{l_m}}, \frac{\frac{1}{1-\gamma} + \frac{1}{1+\gamma}}{1 + \frac{l_m}{u_m} - \frac{1}{du_m}[\frac{1}{1-\gamma} + \frac{1}{1+\gamma}]} \right]. \tag{18}$$

Moreover, due to

$$u_m = \frac{1}{1-\gamma} \sum_{i=1}^{m-1} \frac{\gamma}{\Phi_i} + u_0 \overset{\text{Eqn. (16)}}{\geq} \frac{1}{1-\gamma} \left[ \sum_{i=1}^{m} \frac{\gamma}{\Phi_i} - \frac{1}{d} \right] + u_0 \overset{\text{Eqn. (13)}}{\geq} \frac{1}{1-\gamma} \left[ \frac{2d(1-\gamma^2)}{\gamma^2} - \frac{2}{d} \right] + \frac{2d}{\gamma}. \tag{19}$$

According to lemma. G.2, with probability $(1-\delta)$, we have $\frac{u_m}{l_m} \in [1, 1+8\gamma]$, and hence

$$\lambda(A^\top A) \in \left[ \frac{\frac{1}{1-\gamma} + \frac{1}{1+\gamma}}{1 + 1 + 8\gamma}, \frac{\frac{1}{1-\gamma} + \frac{1}{1+\gamma}}{1 + 1 - \frac{\gamma^2}{[(-2d^2-1)\gamma^2 + d^2\gamma + d^2](1+\gamma)}} \right]. \tag{20}$$

Note that $\gamma \leq \frac{1}{3}$. We have

$$\text{Lower bound of (20)} = \frac{\frac{1}{1-\gamma^2}}{1 + 4\gamma} \geq 1 - 4\gamma,$$

$$\text{Upper bound of (20)} \leq \frac{\frac{2}{1-\gamma^2}}{2 - \frac{\gamma^2}{1+\gamma}\frac{1}{d^2}} \leq \frac{\frac{2}{1-\gamma}}{2(1+\gamma) - \gamma^2} \leq \frac{2(1+\gamma)}{2(1+\gamma) - \gamma^2} \leq 1 + \frac{\gamma^2}{2 + 2\gamma}. \tag{21}$$

Therefore, let $\varepsilon_1 = \min\{4\gamma, \frac{\gamma^2}{2+2\gamma}\}$ yields the first property of lemma 4.3.

**For the second property**, recalling the main text, we aim to prove $\sum_{i=1}^{m} \alpha_i \leq 1 + \varepsilon_2$, and for $\forall i \in [m]$, $\alpha_i \max_k \left\{ \sum_{j=1}^{d} \frac{X_{k,j}^2}{P_{ir(i)}} \right\} \leq \varepsilon_3$.

From Eq. (15), $\frac{\gamma}{\Phi_m} \leq \frac{1}{d} \leq 1$. Therefore,

$$\sum_{i=1}^{m} \alpha_i = \frac{\gamma^2}{2d(1-\gamma^2)} \sum_{i=1}^{m} \frac{\gamma}{\Phi_i} \in \left[ 1, 1 + \frac{\gamma^2}{2d^2(1-\gamma^2)} \right]. \tag{22}$$

Moreover, note that the column of $X$, $\|e_j\| = 1$, and $K_i := \max_k \sum_{j=1}^{d} X_{k,j}^2 \leq \sum_{k \in [n], j \in [d]} X_{k,j}^2 = d$,

$$\alpha_i \max_k \left\{ \sum_{j=1}^{d} \frac{\boldsymbol{X}_{k,j}^2}{P_{ir(i)}} \right\}$$

$$= \frac{\gamma}{\Phi_i} \frac{\gamma^2}{2d(1-\gamma^2)} \max_k \left\{ \frac{\Phi_i}{\boldsymbol{X}_{r(i)}^\top \left[ (u_i I - B_i)^{-1} + (B_i - l_i I)^{-1} \right] \boldsymbol{X}_{r(i)}} \sum_{j=1}^{d} \boldsymbol{X}_{k,j}^2 \right\}.$$

$$\leq \frac{\gamma^3}{2d(1-\gamma^2)} \frac{K_i}{\lambda_{min}\{(u_i I - B_i)^{-1}\} + \lambda_{min}\{(B_i - l_i I)^{-1}\}}$$

$$\leq \frac{\gamma^3}{2d(1-\gamma^2)} \frac{d}{\frac{1}{u_i - l_i} + \frac{1}{u_i - l_i}}$$

$$\leq \frac{\gamma^3}{2d(1-\gamma^2)} \frac{1}{2} * \frac{9d}{\gamma} * d = \frac{\frac{9}{4}d\gamma^2}{1-\gamma^2} := \varepsilon_3,$$

(23)

where the last inequality comes from Claim 18 in Chen & Price (2019).

Hence Algorithm 2 both satisfies the two properties as in Definition 4.1. Combined with lemma G.2, the proof is done.

## C. The proof of Lemma 4.4

*Proof.* Consider each given fixed $\boldsymbol{Y}$, we have that with high probability $(1 - \delta)$,

$$\hat{\boldsymbol{\beta}}_{act} := 2(\tilde{\boldsymbol{X}}^\top \boldsymbol{W} \tilde{\boldsymbol{X}})^{-1} \tilde{\boldsymbol{X}}^\top \boldsymbol{W} \boldsymbol{Y} = (\tilde{\boldsymbol{X}}^\top \boldsymbol{W} \tilde{\boldsymbol{X}})^{-1} \tilde{\boldsymbol{X}}^\top \boldsymbol{W} (\boldsymbol{t} + \boldsymbol{Z} \odot \boldsymbol{\mu}). \tag{24}$$

Hence

$$\begin{aligned}
\left\| \tilde{\boldsymbol{X}} \hat{\boldsymbol{\beta}}_{act} - \boldsymbol{t} \right\|_2^2 &= \left\| \tilde{\boldsymbol{X}} (\tilde{\boldsymbol{X}}^\top \boldsymbol{W} \tilde{\boldsymbol{X}})^{-1} \tilde{\boldsymbol{X}}^\top \boldsymbol{W} (\boldsymbol{t} + \boldsymbol{Z} \odot \boldsymbol{\mu}) - \boldsymbol{t} \right\|_2^2 \\
&\leq \left\| (\tilde{\boldsymbol{X}} (\tilde{\boldsymbol{X}}^\top \boldsymbol{W} \tilde{\boldsymbol{X}})^{-1} \tilde{\boldsymbol{X}}^\top \boldsymbol{W} - I) \boldsymbol{t} \right\|_2^2 + \left\| \tilde{\boldsymbol{X}} (\tilde{\boldsymbol{X}}^\top \boldsymbol{W} \tilde{\boldsymbol{X}})^{-1} \tilde{\boldsymbol{X}}^\top \boldsymbol{W} (\boldsymbol{Z} \odot \boldsymbol{\mu}) \right\|_2^2 \\
&\leq (1 + \epsilon) \min \left\| \tilde{\boldsymbol{X}} \boldsymbol{\beta} - \boldsymbol{t} \right\|_2^2 + \left\| \tilde{\boldsymbol{X}} (\tilde{\boldsymbol{X}}^\top \boldsymbol{W} \tilde{\boldsymbol{X}})^{-1} \tilde{\boldsymbol{X}}^\top \boldsymbol{W} (\boldsymbol{Z} \odot \boldsymbol{\mu}) \right\|_2^2 \\
&\leq (1 + \epsilon) \min \left\| \tilde{\boldsymbol{X}} \boldsymbol{\beta} - \boldsymbol{t} \right\|_2^2 + \sum_{i:w_i \neq 0} \frac{l_i(\sqrt{\boldsymbol{W}} \tilde{\boldsymbol{X}}) \boldsymbol{\mu}_i^2}{\min_i w_i} \\
&\leq (1 + \epsilon) \min \left\| \tilde{\boldsymbol{X}} \boldsymbol{\beta} - \boldsymbol{t} \right\|_2^2 + \frac{d \| \boldsymbol{\mu} \|_\infty^2}{\min_i w_i} \\
&\leq (1 + \epsilon) \min \left\| \tilde{\boldsymbol{X}} \boldsymbol{\beta} - \boldsymbol{t} \right\|_2^2 + \frac{md^2 \| \boldsymbol{\mu} \|_\infty^2 (\max_{i,j} \tilde{\boldsymbol{X}}_{i,j})^2}{1 - \varepsilon_1}.
\end{aligned} \tag{25}$$

Here, the penultimate inequality is deferred to Appendix G.5.

$\square$

## D. The proof of Theorem 4.5

*Proof of Theorem 4.5.* **The unbiasedness** Notice that $\boldsymbol{Y}$ is sampled from Bernoulli sampling for each individual. We conduct the expansion via the actively sampled $S$. We emphasize that the whole randomness of our proposed estimator $\tau_{act}$ is derived from three sources: (i) the random treatment assignment, (ii) the stochastic reweighing vector $\{w_{r(i)}\}_{i \in [m]}$ whose index $(r(i))_{i \in [m]} = S$ is random, and (iii) the uniform sampling process within the random complementary set $\bar{S}$. Then

$$\mathbb{E}(\hat{\tau}_{act} - \tau) := \mathbb{E}_S \mathbb{E}(\hat{\tau}_{act} - \tau | S) \tag{26}$$

Here $S$ is obtained from Algorithm 2. Recalling that $\hat{\tau}_{\bar{S}}$ and $\hat{\tau}_S$ is the unbiased estimation of $\tau$, hence

$$\hat{\tau}_{act} - \tau = -\frac{1}{n}\Big[|\bar{S}|\frac{1}{m}\sum_{i\in\bar{S}_m}\big(\boldsymbol{X}_i - \bar{\boldsymbol{X}}_{\bar{S}_m}\big)^\top + \sum_{i\in S}\big(\boldsymbol{X}_i - \bar{\boldsymbol{X}}_S\big)^\top\Big]\hat{\boldsymbol{\beta}}^{act}. \tag{27}$$

It leads to (fix $S$, $\hat{\boldsymbol{\beta}}^{act}$)

$$\begin{aligned}
\mathbb{E}_S\mathbb{E}[\hat{\tau}_{act} - \tau|S] &= -\frac{1}{mn}\mathbb{E}_S\mathbb{E}\Big[|\bar{S}|\big(\sum_{i\in\bar{S}_m}\big(\boldsymbol{X}_i - \bar{\boldsymbol{X}}_{\bar{S}_m}\big)^\top\hat{\boldsymbol{\beta}}^{act}\big)|S\Big] \\
&= -\frac{1}{mn}\mathbb{E}_S\Big[\mathbb{E}\Big[|\bar{S}|\big(\sum_{i\in\bar{S}_m}\big(\boldsymbol{X}_i - \bar{\boldsymbol{X}}_{\bar{S}_m}\big)^\top\big)|S\Big]\Big]\hat{\boldsymbol{\beta}}^{act}.
\end{aligned} \tag{28}$$

Since $\bar{S}$ is arbitrarily uniformly selected from $[n]/S$, it naturally leads to

$$\mathbb{E}\Big[|\bar{S}|\big(\sum_{i\in\bar{S}_m}\big(\boldsymbol{X}_i - \bar{\boldsymbol{X}}_{\bar{S}_m}\big)^\top\big)|S\Big] = 0. \tag{29}$$

Hence $\mathbb{E}_S\mathbb{E}(\hat{\tau}_{act} - \tau|S) = 0$, and thus $\mathbb{E}(\hat{\tau}_{act} - \tau) = 0$.

**The variance** Conditioning on each random $S, \boldsymbol{Y}$, it leads to (also conditioning on the event where Algorithm 2 could make an $\varepsilon$-error guarantee). Notice that

$$\tau = \frac{|\bar{S}|}{n}\tau_{\bar{S}} + \frac{|S|}{n}\tau_S, \text{ where } \tau_* := \frac{1}{n}\sum_{i\in *}t_i \text{ with } * \in \{\bar{S}, S\}, \bar{S} = n/S. \tag{30}$$

It implies

$$\mathbb{E}[(\hat{\tau}_{act} - \tau)^2|S] = \frac{1}{n^2}\mathbb{E}\big[\big(|\bar{S}|(\hat{\tau}_{act,1} - \tau_{\bar{S}}) + |S|(\hat{\tau}_{act,2} - \tau_S)\big)^2\big] = \text{Error}_1 + \text{Error}_2 + \text{Error}_3, \tag{31}$$

where

$$\begin{aligned}
\text{Error}_1 &:= \frac{|\bar{S}|^2}{n^2}\mathbb{E}\Big[\Big[\Big(\hat{\tau}_{\bar{S}} - \frac{1}{m}\sum_{i\in\bar{S}_m}\big(\boldsymbol{X}_i - \bar{\boldsymbol{X}}_{\bar{S}_m}\big)^\top\hat{\boldsymbol{\beta}}^{act}\Big) - \tau_{\bar{S}}\Big]^2|S\Big], \\
\text{Error}_2 &:= \frac{|S|^2}{n^2}\mathbb{E}\Big[\big(\hat{\tau}_S - \tau_S\big)^2|S\Big], \\
\text{Error}_3 &:= \frac{2|S||\bar{S}|}{n^2}\mathbb{E}\Big[(\hat{\tau}_{act,1} - \tau_{\bar{S}})|S\Big]\mathbb{E}\Big[(\hat{\tau}_{act,2} - \tau_S)|S\Big] = 0.
\end{aligned} \tag{32}$$

**Error$_1$** Notice that $\hat{\tau}_{\bar{S}}$ is derived from the Gram-Schmidt design in Harshaw et al. (2024). We take advantage of the Lemma G.6 as follows:

According to Lemma G.6, given the best coefficient $\widetilde{\boldsymbol{\beta}}^{\bar{S}} = \arg\min\limits_{\boldsymbol{\beta}\in\mathbb{R}^d}\Big[\frac{1}{\phi}\big\|\boldsymbol{X}_{\bar{S}}\boldsymbol{\beta} - \boldsymbol{\mu}_{\bar{S}}\big\|_2^2 + \frac{\zeta^2}{(1-\phi)}\|\boldsymbol{\beta}\|_2^2\Big]$. It implies that when $\hat{\boldsymbol{\beta}}_{act}$ is fixed, we have

$$\begin{aligned}
&\text{Error}_1 \\
&\leq \Big(\frac{|\bar{S}|}{n}\Big)^2\left(\frac{\big\|\boldsymbol{X}_{\bar{S}}\widetilde{\boldsymbol{\beta}}^{\bar{S}} - \boldsymbol{\mu}_{\bar{S}}\big\|_2^2}{m|\bar{S}|\phi} + \frac{\zeta^2\big\|\widetilde{\boldsymbol{\beta}}^{\bar{S}}\big\|_2^2}{m^2(1-\phi)} + \frac{\big\|\big(\overline{\boldsymbol{X}}^{\bar{S}} - \boldsymbol{X}_{\bar{S}}\big)\hat{\boldsymbol{\beta}}_{act} - \big(\bar{\mathbf{t}}^{\bar{S}} - \mathbf{t}_{\bar{S}}\big)\big\|_2^2}{m|\bar{S}|}\right) \\
&\leq \frac{|\bar{S}|}{n}\left[\Big(\frac{1}{mn}\frac{1}{\phi}\Big)\big\|\boldsymbol{X}_{\bar{S}}\widetilde{\boldsymbol{\beta}}^{\bar{S}} - \boldsymbol{\mu}_{\bar{S}}\big\|_2^2 + \frac{1}{m^2}\frac{\zeta^2}{1-\phi}\|\widetilde{\boldsymbol{\beta}}^{\bar{S}}\|_2^2 + \frac{1}{mn}\big\|\big(\overline{\boldsymbol{X}}^{\bar{S}} - \boldsymbol{X}_{\bar{S}}\big)\hat{\boldsymbol{\beta}}_{act} - \big(\bar{\mathbf{t}}^{\bar{S}} - \mathbf{t}_{\bar{S}}\big)\big\|_2^2\right].
\end{aligned} \tag{33}$$

Here the definition of $\bar{\mathbf{t}}^{\bar{S}}, \mathbf{t}_{\bar{S}}$ is analogous to that of $\overline{\boldsymbol{X}}^{\bar{S}}, \boldsymbol{X}_{\bar{S}}$. We additionally set

$$\boldsymbol{\beta}^* = \arg\min\limits_{\boldsymbol{\beta}\in\mathbb{R}^d}\Big[\frac{m}{n}\|\mathbf{X}\boldsymbol{\beta} - \boldsymbol{\mu}\|_2^2 + \frac{\zeta^2}{1-\phi}\|\boldsymbol{\beta}\|_2^2\Big], \widehat{\boldsymbol{\beta}} = \arg\min\limits_{\boldsymbol{\beta}\in\mathbb{R}^d}\big[\|(\boldsymbol{X} - \overline{\boldsymbol{X}})\boldsymbol{\beta} - (\mathbf{t} - \bar{\mathbf{t}})\|_2^2 + \lambda\|\boldsymbol{\beta}\|_2^2\big],$$

then

$$
\begin{aligned}
(33) \leq & \frac{|\bar{S}|}{n} \left[ \left( \frac{1}{mn} \frac{1}{\phi} \right) \| \boldsymbol{X}_{\bar{S}} \boldsymbol{\beta}^* - \boldsymbol{\mu}_{\bar{S}} \|_2^2 + \frac{1}{m^2} \frac{\zeta^2}{1-\phi} \| \boldsymbol{\beta}^* \|_2^2 + \frac{1}{mn} \left\| \left( \overline{\boldsymbol{X}}^{\bar{S}} - \boldsymbol{X}_{\bar{S}} \right) \hat{\boldsymbol{\beta}}_{act} - \left( \overline{\mathbf{t}}^{\bar{S}} - \mathbf{t}_{\bar{S}} \right) \right\|_2^2 \right] \\
\leq & \frac{|\bar{S}|}{n} \left[ \left( \frac{1}{mn} \frac{1}{\phi} \right) \| \boldsymbol{X} \boldsymbol{\beta}^* - \boldsymbol{\mu} \|_2^2 + \frac{1}{m^2} \frac{\zeta^2}{1-\phi} \| \boldsymbol{\beta}^* \|_2^2 + \frac{1}{mn} \left\| \left( \overline{\boldsymbol{X}}^{\bar{S}} - \boldsymbol{X}_{\bar{S}} \right) \hat{\boldsymbol{\beta}}_{act} - \left( \overline{\mathbf{t}}^{\bar{S}} - \mathbf{t}_{\bar{S}} \right) \right\|_2^2 \right].
\end{aligned} \tag{34}
$$

For the third term in the upper bound in (34), according to Algorithm 2, it leads to that with $(1 - \delta)$ probability,

$$
\frac{1}{mn} \left\| \left( \overline{\boldsymbol{X}}^{\bar{S}} - \boldsymbol{X}_{\bar{S}} \right) \hat{\boldsymbol{\beta}}_{act} - \left( \overline{\mathbf{t}}^{\bar{S}} - \mathbf{t}_{\bar{S}} \right) \right\|_2^2 \leq \frac{1}{mn} \left\| \left( \overline{\boldsymbol{X}} - \boldsymbol{X} \right) \hat{\boldsymbol{\beta}}_{act} - \left( \overline{\mathbf{t}} - \mathbf{t} \right) \right\|_2^2. \tag{35}
$$

Finally, we aim to establish the upper bound of the RHS of (35).

**Error$_2$** We could derive the upper bound of the variance of the HT estimator according to Harshaw et al. (2024); Ghadiri et al. (2024):

$$
\text{Error}_2 \leq \frac{|S|^2}{n^2} \frac{\| \sum_{i \in S} (Y_i(0) + Y_i(1)) \|_2^2}{|S|^2} = \frac{\| \sum_{i \in S} (Y_i(0) + Y_i(1)) \|_2^2}{n^2}. \tag{36}
$$

Therefore,

$$
\mathbb{E}[\text{Error}_2] = \mathbb{E}_S \mathbb{E}[\text{Error}_2 | S] \leq (\mathbb{E}|S|) \| Y_i(0) + Y_i(1) \|_\infty^2 / n^2 \leq C d \| \boldsymbol{\mu} \|_\infty^2 / \varepsilon n^2. \tag{37}
$$

Combining with (32), (35), (37), it achieves that with $(1 - \delta)$ probability,

$$
\begin{aligned}
& \mathbb{E}[(\hat{\tau}_{act} - \tau)^2 | S] \\
= & \mathbb{E}_S [\mathbb{E}[(\hat{\tau}_{act} - \tau)^2 | S]] \\
\leq & \frac{C d \| \boldsymbol{\mu} \|_\infty^2}{\varepsilon n^2} + \min_{\beta} \left[ \left( \frac{1}{mn} \frac{1}{\phi} \right) \| \boldsymbol{X} \boldsymbol{\beta} - \boldsymbol{\mu} \|_2^2 + \frac{1}{m^2} \frac{\zeta^2}{1-\phi} \| \boldsymbol{\beta} \|_2^2 \right] + \frac{1+\varepsilon}{mn} \min_{\boldsymbol{\beta}} \left\| \tilde{\boldsymbol{X}} \boldsymbol{\beta} - \mathbf{t} \right\|_2^2 + \frac{d \| \boldsymbol{\mu} \|_\infty^2}{\underline{w} mn}.
\end{aligned} \tag{38}
$$

$\square$

# E. The Proof of lower bound

*The sketch of proof.* We consider the following constructions: $\tau(\boldsymbol{X}_i) = L(\boldsymbol{X}_i) + \mu$. Here $L(\cdot)$ is the pre-fixed function, selected from the linear family $\mathcal{L}$ satisfying $\| \mathcal{L} \|_D = 1$[5]. $\mu$ is the i.i.d. Gaussian noise satisfying $\mu \sim N(0, \frac{1}{\varepsilon})$. $D$ is considered as a uniform distribution under the $d$ dimensional Euclidean space. We consider that $\boldsymbol{X}$ is sampled from $D$, and then we construct the potential outcome $Y_i(\boldsymbol{X}_i)$. On this basis, we extract a specific subset of $\mathscr{L} \subseteq \mathcal{L}$:

**Lemma E.1.** *(Sparsity of $\mathcal{L}$) Chen & Price (2019) $\exists$ a subset $\{ L_1, L_2, ... L_s \} =: \mathscr{L} \subseteq \mathcal{L}$ with $s \geq 2^{0.7d}$, such that $\| L_i \|_D = 1, \| L_i \|_\infty \leq 1$. Moreover, $\| L_i - L_j \|_D \geq 0.2$.*

*The sketch of proof of Lemma E.1.* We provide the construction as follows: In this procedure, we start with $n = 0$ and let $\mathscr{L}$ be all functions mapping $[d]$ to $\{\pm 1\}$. While $\mathscr{L}$ is nonempty, we pick any function $h$ in $\mathscr{L}$, remove from $U$ all functions within distance 0.2 of $h$, increment $n$ by one, and name that function $f_n$. After no functions remain, we return the set $\mathscr{L} = \{ L_1, \ldots, L_s \}$. In the initialization step, $|U| = 2^d$, then in the above process, according to Chen & Price (2019), we claim this process remove $C_d^{0.01d} \leq 2^{0.081d}$ each round, and hence it follows that $s \geq 2^d / 2^{0.081d} = 2^{1.919d}$.

We consider the mutual information between the selected linear function family and our algorithm's output, i.e., estimator, $\hat{\tau}$. On the one hand, we provide the lower bound of such mutual information by Fano's inequality. Specifically, it leads to

$$
I(L_j; \hat{\tau}) = H(L_j) - H(L_j \mid \hat{\tau}) \geq log(s) - 1 - log(s-1)/4 \geq 1.43. \tag{39}
$$

---

[5]We define $\| g(\cdot) \|_L^2 := \mathbb{E}_{x \sim D} |g(x)|^2$.

On the other hand, we aim to provide the upper bound according to the Shannon-Hartley theorem:

$$
\begin{aligned}
I(\hat{\tau}; L_j) \leq & I((\tau_1, \tau_2, ...\tau_s); L_j) \\
= & \sum_{i \in [s]} I(\tau_i; L_j(\boldsymbol{X}_i) \mid \tau_1, ...\tau_{i-1}) \\
\leq & \frac{s}{2} \log \left( 1 + \frac{\max_{L \in \mathscr{L}} \left[ L(\boldsymbol{X}_i)^2 \right]}{1/\varepsilon} \right) \\
= & \frac{s}{2} \log(1 + \varepsilon) \leq \frac{s\varepsilon}{2}.
\end{aligned}
\tag{40}
$$

Combined with (39) and (40), it implies that

$$
s \geq \frac{2.86d}{\varepsilon}.
\tag{41}
$$

The result follows. $\qquad\square$

## F. The Proof of Proposition 6.3

*Proof.* Notice that

$$
\hat{\tau}_{\bar{S}} := \frac{1}{|\bar{S}'_m|} \sum_{i \in \bar{S}_m`} (Y_i - \boldsymbol{X}_i^{\mathbf{G}} \boldsymbol{\beta}) \left( \frac{\mathbf{1}\left[ E_{(i,1)} \right]}{\Pr\left( E_{(i,1)} \right)} - \frac{\mathbf{1}\left[ E_{(i,0)} \right]}{\Pr\left( E_{(i,0)} \right)} \right).
$$

On this basis, it leads to $\hat{\tau}_{act} - \tau = \frac{|\bar{S}|}{n}(\hat{\tau}_{act,1} - \tau) + \frac{|S|}{n}(\hat{\tau}_{act,2} - \tau)$, then it follows that

$$
\mathbb{E}[(\hat{\tau}_{act} - \tau)^2] \leq \frac{2|\bar{S}|^2}{n^2}(\hat{\tau}_{act,1} - \tau)^2 + \frac{2|S|^2}{n^2}(\hat{\tau}_{act,2} - \tau)^2
$$

The first part could be bounded by $\frac{2d^2}{\varepsilon^2 n^2} C$, where $C$ is a constant. For the second part, according to Kandiros et al. (2024), it leads to

$$
\frac{|\bar{S}|^2}{n^2} \frac{25\lambda(\mathcal{H})}{|\bar{S}|^2} \sum_{j=0,1} \sum_{i \in |\bar{S}|} (Y_i(e_j) - \boldsymbol{X}_i^{\mathbf{G}} \boldsymbol{\beta}^{act})^2.
$$

According to a similar analogy in Theorem 4.4, it follows that $\|\boldsymbol{X}_i^{\mathbf{G}} \boldsymbol{\beta}^{act} - \frac{1}{2}t_i\| \leq$

$$
(1 + \epsilon) \min \left\| \boldsymbol{X}^{\mathbf{G}} \boldsymbol{\beta} - \frac{1}{2}\boldsymbol{t} \right\|_2^2 + \frac{md^2 \|\boldsymbol{\mu}\|_\infty^2 \|\boldsymbol{X}\|_\infty^2}{1 - \varepsilon_1}.
$$

$\qquad\square$

## G. Auxiliary lemmas

**Lemma G.1** (Random vector regression (Ghadiri et al., 2024)). *. Let* $\mathbf{y}^{(0)}, \mathbf{y}^{(1)} \in \mathbb{R}^n$ *and* $\mathbf{y} \in \mathbb{R}^n$ *be a random vector such that for each* $i \in [n], y_i$ *is independently and with equal probability is either equal to* $y_i^{(0)}$ *or* $y_i^{(1)}$. *Moreover, let* $\mathbf{b}^* = \arg\min_{\mathbf{b}} \|\mathbf{X}\mathbf{b} - \mathbf{y}\|_2^2$. *Let* $\boldsymbol{\mu} := \mathbf{y}^{(1)} + \mathbf{y}^{(0)}$. *Then*

$$
\mathbb{E}\left[ \|2\mathbf{X}\mathbf{b}^* - \boldsymbol{\mu}\|_2^2 \right] \leq d \left\| \mathbf{y}^{(1)} - \mathbf{y}^{(0)} \right\|_\infty^2 + \min_{\mathbf{b}} \|\mathbf{X}\mathbf{b} - \boldsymbol{\mu}\|_2^2
$$

**Lemma G.2.** *(Lee & Sun (2018); Chen & Price (2019)) There exists a constant* $C$, *such that with at least probability* $(1 - \delta)$, *Algorithm. 2 can be terminated by* $m \leq C\frac{d}{\gamma^2} = O(\frac{d}{\varepsilon})$ *iterations and* $\frac{u_m}{l_m} \in [1, 1 + 8\gamma]$.

**Lemma G.3.** *(Lee & Sun (2018); Batson et al. (2009)) The eigenvalues of* $B_i$ *in Algorithm. 2 is bounded, namely* $\lambda(B_i) \in (l_i, u_i)$.

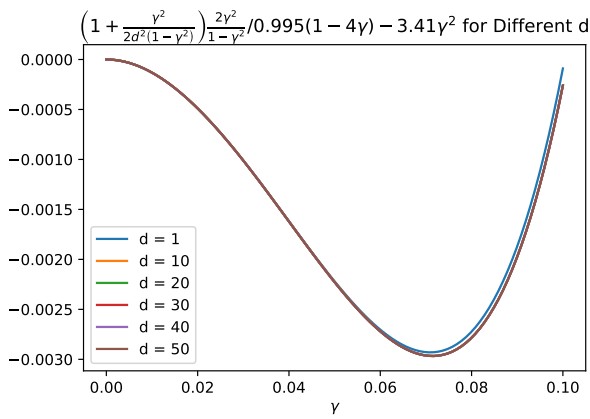

*Figure 2.* Illustration for Lemma G.4.

**Lemma G.4.**

$$\forall \gamma \in [0, 0.1], (1 + \frac{\gamma^2}{2d^2(1-\gamma^2)}) \frac{2\gamma^2}{(1-\gamma^2)(1-\delta)(1-4\gamma)} \leq 3.41\gamma^2. \tag{42}$$

**Lemma G.5.**

$$w_{min} \geq \frac{1 - \varepsilon_1}{d \cdot a^2}.$$

*Proof.* The matrix $\boldsymbol{A} \in \mathbb{R}^{m \times d}$ now has elements: $A(i,j) = \sqrt{w_i} X_{i,j}$, where $\boldsymbol{X}$ is an $m \times d$ matrix with bounded entries: $X_{i,j} \in [a, b]$, for some constants $a, b > 0$.

We aim to find the minimum $w_i$ that ensures the eigenvalues of $\boldsymbol{A}^\top \boldsymbol{A}$ satisfy: $\lambda(\boldsymbol{A}^\top \boldsymbol{A}) \in [1 - \varepsilon_1, 1 + \varepsilon_1]$, with at least $1 - \delta$ probability.

**Step 1: Matrix $\boldsymbol{A}$ and its structure** The matrix $\boldsymbol{A}$ now has entries: $A(i,j) = \sqrt{w_i} X_{i,j}$.

The matrix product $\boldsymbol{A}^\top \boldsymbol{A}$ is given by: $\boldsymbol{A}^\top \boldsymbol{A} = \sum_{i=1}^m w_i \boldsymbol{x}_i \boldsymbol{x}_i^\top$, where $\boldsymbol{x}_i$ is the $i$-th row of $\boldsymbol{X}$ treated as a vector in $\mathbb{R}^d$. Thus, $\boldsymbol{A}^\top \boldsymbol{A}$ is a weighted sum of the outer products of the rows of $\boldsymbol{X}$.

**Step 2: Eigenvalue properties of $\boldsymbol{A}^\top \boldsymbol{A}$** The eigenvalues of $\boldsymbol{A}^\top \boldsymbol{A}$ depend on the row vectors $\boldsymbol{x}_i$ and their weights $w_i$. Specifically: $\lambda(\boldsymbol{A}^\top \boldsymbol{A}) = \lambda \left( \sum_{i=1}^m w_i \boldsymbol{x}_i \boldsymbol{x}_i^\top \right)$.

Using the Rayleigh quotient for eigenvalues: $\lambda_{\max}(\boldsymbol{A}^\top \boldsymbol{A}) = \sup_{\|v\|_2 = 1} \left\| \boldsymbol{A}^\top \boldsymbol{A} v \right\|_2^2$.

This eigenvalue depends on the magnitude and alignment of the weighted row vectors $\sqrt{w_i} \boldsymbol{x}_i$. If $\|\boldsymbol{x}_i\|_2^2$ is bounded, this affects the scaling.

**Step 3: Row norm of $\boldsymbol{x}_i$** The norm of each row $\boldsymbol{x}_i$ is bounded because $X_{i,j} \in [a, b]$. The maximum and minimum row norms satisfy: $\|\boldsymbol{X}_i\|_2^2 \in [d \cdot a^2, d \cdot b^2]$, where: - $d \cdot a^2$: The minimum row norm when all $X_{i,j} = a$. - $d \cdot b^2$: The maximum row norm when all $X_{i,j} = b$.

**Step 4: Bounds on $\boldsymbol{A}^\top \boldsymbol{A}$** The matrix $\boldsymbol{A}^\top \boldsymbol{A}$ can be written as: $\boldsymbol{A}^\top \boldsymbol{A} = \sum_{i=1}^m w_i \boldsymbol{x}_i \boldsymbol{x}_i^\top$.

The eigenvalues of $\boldsymbol{A}^\top \boldsymbol{A}$ are bounded as: $\lambda_{\min}(\boldsymbol{A}^\top \boldsymbol{A}) \geq \min_i \left( w_i \|\boldsymbol{x}_i\|_2^2 \right)$, $\lambda_{\max}(\boldsymbol{A}^\top \boldsymbol{A}) \leq \sum_{i=1}^m w_i \|\boldsymbol{x}_i\|_2^2$.

Substituting the bounds on $\|\boldsymbol{x}_i\|_2^2$: $\lambda_{\min}(\boldsymbol{A}^\top \boldsymbol{A}) \geq \min_i \left( w_i \cdot d \cdot a^2 \right)$, $\lambda_{\max}(\boldsymbol{A}^\top \boldsymbol{A}) \leq \sum_{i=1}^m w_i \cdot d \cdot b^2$.

**Step 5: Ensuring $\lambda(\boldsymbol{A}^\top \boldsymbol{A}) \in [1 - \varepsilon_1, 1 + \varepsilon_1]$** To ensure the eigenvalues lie within $[1 - \varepsilon_1, 1 + \varepsilon_1]$, we require: $1 - \varepsilon_1 \leq \lambda_{\min}(\boldsymbol{A}^\top \boldsymbol{A})$ and $\lambda_{\max}(\boldsymbol{A}^\top \boldsymbol{A}) \leq 1 + \varepsilon_1$.

Using the bounds: 1. For $\lambda_{\min}(\boldsymbol{A}^\top \boldsymbol{A})$: $1 - \varepsilon_1 \leq \min_i \left( w_i \cdot d \cdot a^2 \right)$. This gives: $w_{\min} \geq \frac{1-\varepsilon_1}{d \cdot a^2}$.

$\square$

---

**Algorithm 4** Conflict-Graph-Design (`CGD`, following Kandiros et al. (2024))

---

**Require:** Importance ordering $\pi$, maximum eigenvalue $\lambda(\mathcal{H})$, and conflict graph $\mathcal{H}$.

1: Set sampling parameter $r = 2$
2: Sample desired exposure variables $U_1, \ldots, U_n$ independently and identically as $U_i \leftarrow$

$$\begin{cases} e_1 & \text{with probability } \frac{1}{r \cdot 2\lambda(\mathcal{H})} \\ e_0 & \text{with probability } \frac{1}{r \cdot 2\lambda(\mathcal{H})} \\ * & \text{with probability } 1 - \frac{1}{r \cdot \lambda(\mathcal{H})} \end{cases}$$

3: Initialize the intervention vector $Z \leftarrow \mathbf{0}$.
4: **while** $i = 1, 2, \ldots n$ **do**
5:     **if** $U_i \in \{e_1, e_0\}$ and $U_j = *$ for all $j \in \mathcal{N}_b^\pi(i)$ **then**
6:         Update intervention vector: set $Z_j = \mathbf{Z}_{i,(k)}(j)$ for all $j \in \tilde{N}(i)$ where $U_i = e_k$.
7:     **end if**
8: **end while**

**Ensure:** Random intervention $Z \in \mathcal{Z} = \{0,1\}^n$.

---

| Method | Variance | Sample complexity |
|---|---|---|
| HT estimator | $\frac{\|\boldsymbol{\mu}\|_2^2}{mn} + \frac{S_t^2}{m}\left(1 - \frac{m}{n}\right)$ | $m$ |
| GSW design (Harshaw et al., 2024) | $\frac{1}{m^2}\min_{b\in\mathbb{R}^d}\left[\frac{m}{n}\cdot\frac{1}{\phi}\|\mathbf{X}\mathbf{b} - \boldsymbol{\mu}\|_2^2 + \frac{1}{1-\phi}\zeta^2\|\mathbf{b}\|_2^2\right] + \frac{S_t^2}{m}\left(1 - \frac{m}{n}\right)$ | $m$ |
| RAHT (Ghadiri et al., 2024) | $\frac{1}{mn}\frac{1}{\phi}\|\mathbf{X}\mathbf{b}^* - \boldsymbol{\mu}\|_2^2 + \frac{1}{m^2}\frac{\zeta^2}{(1-\phi)}\|\mathbf{b}^*\|_2^2 + \frac{100d\cdot\log(d/\delta)}{n^2\epsilon^2}\|\boldsymbol{\mu}\|_\infty^2$ $+(1+\epsilon)\cdot\left(\frac{1}{m}\|\boldsymbol{\mu}\|_\infty^2 + \frac{1}{mn}\|(\mathbf{X}-\overline{\mathbf{X}})\widehat{\mathbf{b}} - (\mathbf{t}-\overline{\mathbf{t}})\|_2^2 + \frac{\lambda}{mn}\|\widehat{\mathbf{b}}\|_2^2\right)$ | $m + \mathcal{O}(d\log(d)/\varepsilon^2)$ |
| Leverage score (Addanki et al., 2022) | (mainly consider the absolute bias, and relies on Gaussian noise assumption) | $\mathcal{O}(d\log(d) + d/\varepsilon)$ |
| **Ours** | $\text{error}_{\overline{\mathbf{S}}} + \text{error}_{\overline{\mathbf{S}},\text{GSW}} + \text{error}_{\overline{\mathbf{S}},\text{ITE}}$ | $\mathcal{O}(d/\varepsilon + m')$ |

*Table 4.* Comparison of different active sampling for estimating ATE inherited from Ghadiri et al. (2024). $0 < \phi < 1$, $0 < \epsilon < 1$ are parameters in `GSW` design. Moreover, $\zeta$ is the maximum $\ell_2$ norm over rows of $\mathbf{X}$. $S_t$ denotes the standard variance of $\boldsymbol{t}$

**Lemma G.6.** *(Harshaw et al. (2024); Ghadiri et al. (2024)) Let $\widehat{\tau}_S$ be an estimate obtained by the Horvitz-Thompson estimator on the Gram-Schmidt walk design. Let $\widehat{\mathbf{b}}$ be a fixed/preassigned vector and $0 < \phi < 1$. Then, the partial observation regression-adjusted Horvitz-Thompson estimator with $m$ samples is an unbiased estimator with variance of:*

$$\mathbb{E}\left[(\widehat{\tau} - \tau)^2\right] \leq \frac{1}{mn}\frac{1}{\phi}\|\boldsymbol{X}\mathbf{b}^* - \boldsymbol{\mu}\|_2^2 \quad + \frac{1}{m^2}\frac{\zeta^2}{(1-\phi)}\|\mathbf{b}^*\|_2^2 \quad + \frac{1}{mn}\left\|(\boldsymbol{X} - \overline{\boldsymbol{X}})^\top\widehat{\mathbf{b}} - (\mathbf{t} - \overline{\mathbf{t}})\right\|_2^2, \tag{43}$$

*where*

$$\mathbf{b}^* = \arg\min_{\mathbf{b}\in\mathbb{R}^d}\left[\frac{1}{\phi}\|\boldsymbol{X}\mathbf{b} - \boldsymbol{\mu}\|_2^2 + \frac{\zeta^2}{(1-\phi)}\|\mathbf{b}\|_2^2\right]. \tag{44}$$

**Lemma G.7.** $C_d^{0.01d} \leq 2^{0.081d}$.

*Proof.* To establish a smaller constant $c$ for the inequality $\binom{d}{0.01d} \leq 2^{cd}$, we use an upper bound involving the binary entropy function. A well-known result states that for $n \in \mathbb{N}$ and $0 < p < 1$, the binomial coefficient $\binom{n}{pn}$ is bounded above by $2^{nH(p)}$, where $H(p) = -p\log_2(p) - (1-p)\log_2(1-p)$ is the binary entropy. In our case, with $p = 0.01$, we compute $H(0.01) \approx 0.0807$. Consequently, for sufficiently large $d$, $\binom{d}{0.01d} \leq 2^{dH(0.01)} \approx 2^{0.0807d}$. Hence, by choosing any $c \geq 0.0807$, we guarantee $\binom{d}{0.01d} \leq 2^{cd}$. $\qquad\square$

## H. Auxiliary algorithms

Seen in Algorithm 4 and Algorithm 5.

## I. Comparison with previous bounds

Seen in Table 4.

---

**Algorithm 5** `GSW` design (following Harshaw et al. (2024))

---

**Require:** Matrix $\boldsymbol{X} \in \mathbb{R}^{n \times d}, 0 < \phi < 1$, vector $\mathbf{p} \in \mathbb{R}^n$.

Initialize an index $j \leftarrow 1$. Select an initial pivot unit $k$ uniformly at random from $[n]$. Set $\zeta \leftarrow \max_{i \in [n]} \|\boldsymbol{X}_{i:}\|_2$. Let

$\mathbf{B} \in \mathbb{R}^{n \times (n+d)}$ be a matrix such that $\mathbf{B}_{i:} := \left[ \begin{array}{c} \sqrt{\phi} \cdot \mathbf{e}_i \\ \zeta^{-1} \sqrt{1-\phi} \cdot \boldsymbol{X}_{i:} \end{array} \right]$, where $\mathbf{e}_i$ is the $i$, the basis vector of dimension $n$.

    **while** $\mathbf{z}^{(j)} \notin \{-1, +1\}^n$ **do**

      Create the set $S \leftarrow \left\{ i \in [n] : \left| z_i^{(j)} \right| < 1 \right\}$.

      **if** $k \notin S$ **then**

        Select a new pivot $k$ from $S$ uniformly at random.

      **end if**

      Compute a step direction as $\mathbf{u}^{(j)} \leftarrow \arg\min_{\mathbf{u}} \{ \|\mathbf{B}\mathbf{u}\|_2 : u_i = 0 \text{ for all } i \notin S, u_k = 1 \}$.

      Let $\Delta = \left\{ \delta \in \mathbb{R} : \mathbf{z}^{(j)} + \delta \cdot \mathbf{u}^{(j)} \in [-1, 1]^n \right\}$.

      Set $\delta^+ \leftarrow |\max \Delta|$ and $\delta^- \leftarrow |\min \Delta|$. Pick a random step size $\delta_j$ that is equal to $\delta^+$ with probability $\delta^- / (\delta^+ + \delta^-)$, and is equal to $-\delta^-$ with probability $\delta^+ / (\delta^+ + \delta^-)$.

      Update fractional assignment: $\mathbf{z}^{(j+1)} \leftarrow \mathbf{z}^{(j)} + \delta_j \mathbf{u}^{(j)}$.

      Increment the index $j \leftarrow j + 1$.

    **end while**

**Ensure:** Return the assignment vector $\mathbf{z}^{(j)} \in \{-1, +1\}^n$.

---

