# OpenReview forum: "Active Treatment Effect Estimation via Limited Samples"
_ICML.cc/2025/Conference — ICML 2025 poster_

### Official Review · Reviewer_fVjN · 2025-03-11

**Overall Recommendation:** 4

**Summary:**

The paper proposes a new active learning strategy for experimental design and an accompanying ATE estimator, derived from literature on estimators with finite sample guarantees. This is especially relevant to cases where experimental sampling must be constrained due to cost or other concerns. Additionally, the paper briefly discusses an extension to scenarios where SUTVA is violated.

**Claims And Evidence:**

The key contributions stated in the introduction are all supported by clear evidence.

**Essential References Not Discussed:**

Nothing notable is omitted from my knowledge.

**Experimental Designs Or Analyses:**

For the primary proposed algorithm, the experimental section was sound and the baselines were consistent with other papers in this area.

However, there is no experimental section for the SUTVA relaxed CGAS method. This does not impact the primary argument of the paper, but perhaps hints that the Section 6 material is better saved for an expanded sequel with more analysis.

One suggestion: The difference in sample complexity between the proposed RWAS method and the previously proposed RAHT method is primarily a function of the feature space size $d$, so I suspect further experiments investigating empirical performance differences as $d$ varies may be interesting. Further reinforcing this, in Table 2, the Twins data set shows the largest different between RWAS and RAHT and also happens to have the largest feature space.

**Methods And Evaluation Criteria:**

The proposed methods make sense for the problem.

**Other Comments Or Suggestions:**

I think there may be notational issues with the index $j$ in definition 4.1. Initially we extract the sample with index $r(i) = j$, so presumably $X_{r(i)} = X_j$, which is used to create $P_{ij}$. In a following sentence where $A$ is defined, $j$ now denotes the $j$th coordinate of $X_i$, and we confusingly have $X_{r(i),j}$ despite previously defining $r(i) = j$ a few sentences prior.

in the section 6 header, 'assumption' is misspelled as assupmtion

**Other Strengths And Weaknesses:**

The paper does a good job of presenting the problem and discussing the solution. It was relatively easy to follow their arguments and evaluation.

There are some minor notational issues, and while section 6 appears to be quite valuable, it does not appear to have the space it deserves, including empirical validation.

**Questions For Authors:**

In Table 1, the sample complexity for Addanki et al. is stated as $O(d log(d) + d/\epsilon)$, which is the sample complexity for leverage sampling. However, only the proposed ITE estimator in that paper relies on leverage sampling. Their proposed ATE estimator instead recursively partitions the data using the GSW algorithm from Harshaw et al., so presumably the sample complexity would be different. Am I mistaken?

**Relation To Broader Scientific Literature:**

The paper adapts the proposed method from Ghadiri et al to include a more efficient sampling strategy, derived from the more theoretical work of Chen and Price. The proposed estimator shows improved sample efficiency both theoretically and empirically.

Within the much broader literature, experiments with limited sample sizes are quite prevalent due to a number of reasons, including high costs or ethical concerns. The proposed methodology seems like a potentially valuable solution to this common scenario.

**Theoretical Claims:**

I only briefly reviewed the proofs, and do not have any immediate concerns about correctness.

The proof of Lemma 4.4 borrows notation from Ghadiri et al without defining it or referencing that paper, making it hard to follow without previously reading the other paper. In particular, $2y = t + Z \bigodot \mu$. Also, it does not appear that $\mu$ is formally defined in the main body of the text, but is presumably consistent with the definition from Ghadiri et al.

---

> ### Author Rebuttal · Authors · 2025-04-01
>
> Thanks for your advice! Here is our response point by point:
>
> > **Q1: The proof of Lemma 4.4 borrows notation from Ghadiri et al. without defining it or referencing that paper, making it hard to follow without previously reading the other paper. In particular, $2 y=t+Z \bigodot \mu$. Also, it does not appear that $\mu$ is formally defined in the main body of the text, but is presumably consistent with the definition from Ghadiri et al.**
>
> Thanks for the careful review and wonderful comment! Here $\mu$ indicates that the vector composes of $Y_i(0)+Y_i(1)$. and $\bigodot$ denots the Hadamard product. We agree that, even though we had previously stated that these local notations were adopted from Ghadiri et al., it is important to make this explicit again in the main text or appendix. We have revised the manuscript accordingly with our utmost care and sincerity, which will be presented with full rigor and clarity in the final camera-ready version.
>
> > **Q2: Add network experiment**
>
> Thanks for your advice. First, we have conducted large-scale experiments across a range of values for both $d$ and $n$, as shown in  `RW kg3L`.
> Second, we have fully implemented your suggestion and carried out additional experiments under settings with network interference. Specifically, we followed both the simulation setup and the real-world dataset configuration used in the work by Lu et al. [1]. We replicate their basic settings wholly. We refer reader for the experimental results in the link https://anonymous.4open.science/r/Causal-Active-Sample-B0C5/README.md.
>
> > **Q3: Expeiremnt focuses on d.**
>
> Refer to `RW kg3L`.
>
> > **Q4: Notation issue**
> Thanks for your comment! We will fix this notational issue by choosing another symbol to denote the row of the vector or matrix to avoid potential incomprehension.
>
>
> >**Q5: The proposed ATE estimator in Addanki et al. instead recursively partitions the data using the GSW algorithm from Harshaw et al., so presumably the sample complexity would be different.**
>
>
> Thanks for your comment. The sample complexity for Addanki et al is $O(dlog(d)+d/\epsilon)$ for the ITE estimation; for ATE estimation, in their excellent paper, the complexity could be controlled by "ATE estimation. For ATE estimation we give a randomized algorithm that selects at most $s$ individuals for treatment/control assignment and obtains an error of $\widetilde{O}\left(\sigma / \sqrt{s}+\left(\left\|\boldsymbol{\beta}^1\right\|+\left\|\boldsymbol{\beta}^0\right\|\right) / s\right)$," We will revise it in the final version. Thanks for your wonderful comment!
>
> ----
> [1] Adjusting auxiliary variables under approximate neighborhood interference, X Lu, Y Wang, Z Zhang, arXiv preprint arXiv:2411.19789.

---

### Official Review · Reviewer_RneH · 2025-03-12

**Overall Recommendation:** 4

**Summary:**

Experimental design for estimating treatment effects does not generally have strong finite-sample guarantees, especially as the dimensionality of the covariates grows. Recent works implement experimental design based on leverage scores. This work proposes an alternative approach called IRD, which helps achieve a sample complexity for the estimation error that is linear in the covariate dimensionality. The method is validated with a variety of standard semi-synthetic experiments.

**Update after rebuttal**: after considering the additional results provided, I have decided to increase my score.

**Claims And Evidence:**

The contribution is clear and the method is sufficiently demonstrated on a variety of standard benchmarks. It would be helpful to see results that specifically highlight the finite-sample guarantee as a function covariate dimensionality. Without this, there is little intuition on the supposed performance benefits.

**Essential References Not Discussed:**

References are sufficient.

**Experimental Designs Or Analyses:**

The paper presents numerous standard benchmarks in treatment-effect estimation like IHDP and the Boston dataset, for different subsample sizes.

**Methods And Evaluation Criteria:**

The core novelties of this method like the adaptation of IRD are never fully described. Section 4.1 gives some intuition and a broad overview of the goals for the proposed method, but the steps in Algorithms 1 and 2 are not justified in detail. In my view, the authors need to put more effort into clearly explaining the motivation behind the new algorithm, beyond referencing recent works that they are building upon.

**Other Comments Or Suggestions:**

Some of the language is inappropriate for a venue like ICML. For instance, in the beginning of Section 4, the authors use exaggerated adjectives like "outstanding" and "superior" without reference to specific claims or objectives.

**Other Strengths And Weaknesses:**

The problem of active sampling for treatment-effect estimation with high-dimensional covariates is clearly significant. The solution appears to have clear benefits over other recent works. It would be very helpful to better describe the method so that readers can understand the key contributions.

**Questions For Authors:**

1. Specifically what role do partitioning and subsampling play in the proposed method?

2. Does this method easily extend to multiple treatments?

**Relation To Broader Scientific Literature:**

The contributions are clearly stated in relation to recent works like those of Ghadiri et al. and Harshaw et al.

**Theoretical Claims:**

I did not check the lengthy appendices that contain all of the proofs. However, I took a brief look and there are no obvious errors.

---

> ### Author Rebuttal · Authors · 2025-04-01
>
> Thanks for your review and comments! Here are the responses to all of your questions.
> > **Q1: It would be helpful to see results as a function of covariate dimensionality.**
>
> Thanks for your suggestion! We provide the following supplementary experiments on the performance differences as d varies with $n=1000$ samples. The advantage of our RWAS algorithm over traditional CRA/GSW methods is especially apparent when the sample dimension d is large, as a result of the optimal sample complexity in active sampling process.
> | d     | HT         | Hajek      | CRA         | GSW         | RAHT        | SC          | 4-Vectors   | RWAS (Ours)   |
> |-------|------------|------------|-------------|-------------|-------------|-------------|-------------|---------------|
> | 10    | 1.98 (0.78) | 1.27 (0.53) | 0.98 (0.40) | **0.93 (0.38)** | 1.08 (0.47) | 1.13 (0.65) | 1.15 (0.53) | 1.08 (0.46)    |
> | 20    | 3.67 (0.88) | 2.88 (0.57) | **2.10 (0.49)** | 2.28 (0.70) | 2.41 (0.59) | 2.50 (0.73) | 2.46 (0.62) | 2.19 (0.55)    |
> | 50    | 1.14 (0.74) | 1.03 (0.50) | 1.02 (0.39) | 0.80 (0.64) | 0.80 (0.56) | 0.80 (0.68) | 0.92 (0.61) | **0.71 (0.47)** |
> | 100   | 1.82 (0.95) | 1.62 (0.86) | 2.22 (0.53) | 1.78 (0.87) | 1.53 (0.83) | 1.73 (0.90) | 1.80 (0.85) | **1.51 (0.82)** |
>
> > **Q2: Further description on the core novelties of this method like the adaptation of IRD.**
>
> Thanks for your concern. Please see the response to `RW PucS`.
>
> > **Q3: Revision on the word like outstanding**
>
> Thank you for this helpful note. In response, we have carefully revised the wording in the beginning of Section 4 to remove exaggerated adjectives such as "outstanding" and "superior". We now use more neutral, objective language and anchor our claims directly to specific empirical results and theoretical guarantees. This revision aims to improve clarity and expression throughout the manuscript. We sincerely appreciate your feedback on this matter.
>
> >**Q4: How to emphasize the role of partitioning & subsampling**
>
> Thanks for your concern! The overall target of partitioning and subsampling is to find efficient causal effect estimators with limited cost on sample collection and computation, but focusing on different components.
>
> * Subsampling finds the most representative samples in terms of covariate distribution by selecting and up-weighting ``influential'' directions in the data.
>
> * Partitioning seeks for well-balanced assignments and provide unbiased effect estimators with controllable expected error.
>
> In Algorithm 1, partitioning and subsampling are deployed in two disjoint subset of the entire samples, one of which applies GSW with regression adjustment, and the other applies IRD. Through weighting and averaging, we obtain an unbiased estimate with the total sample size being O(d).
>
> Besides, the key idea is to improve the MSE through well-balanced assignments and representative samples. As stated in line 65, two types of techniques are applied to achieve these goals: partitioning and subsampling. The IRD algorithm selects the representative samples to ensure a controllable estimation error. At the same time, the GSW design is applied to achieve balanced assignments by partitioning the samples into treatment and control groups. This motivates the following thought, serving as another line of motivation:
>
> * We can use O(1) samples for the GSW regression-adjusted estimator.
> * However, learning the regression coefficient requires O(d) samples.
> * Therefore, we divide the data into two disjoint sets, one of which applies GSW with regression adjustment, and the other applies IRD, using O(d) samples to learn the regression coefficient from the first set.
> * These two sets each construct an estimator, and after weighting and averaging, we obtain an unbiased estimate, with the total sample size being O(d). Note that using either set alone as an estimator may be biased because we actively select individuals corresponding to X with good representational properties for the estimation. This group is not independent and is randomly selected from the entire sample.
>
> > **Q5: Does this method easily extend to multiple treatments?**
>
> Thanks for your question. It is not complicated to propose a solution for multiple-treatment extension. For example, the estimator on the treatment effects between two treatment levels $Z=z_1$ and $Z=z_2$ can be constructed through our method restricting treatment allocation on $\{z_1,z_2\}$ as long as the positivity assumption $P(Z_i=z)>0$ holds for targeted levels. On the other hand, it is possible to find methods to compare different treatment pairs through one sampling process, or combine multiple processes to further increase efficiency, which requires further research. It might be highly nono-trivial due to the proof related to the martingale in the GSW design.

---

### Official Review · Reviewer_PucS · 2025-03-16

**Overall Recommendation:** 3

**Summary:**

The authors considered the problem of estimating the causal effect in an active sampling framework. In particular, they proposed a method called RWAS which attains the sample query of $O(d/\epsilon)$ to achieve $\epsilon$-approximation error where $d$ is the number of covariates. Moreover, they also provided a lower bound, showing the optimality of the proposed method up to some constant. They performed experiments on synthetic and real datasets showing that the proposed method has a better performance compared to baseline methods.

### Update after rebuttal
I checked the proof of the lower bound given in the rebuttal and it looks correct to me. I suggested adding these explanations to the paper. I adjusted my score. However, I still believe that the paper is not well-written. Therefore, I could not recommend a clear acceptance.

**Claims And Evidence:**

The paper is a theoretical work and proofs are provided for theorems or lemmas in the paper. However, the paper is not well-written and it is hard to follow some parts of it For instance, some terms or definitions are not defined. Moreover, it is often assumed that the reader is completely knowledgable about all aspects of the problem. However, the authors should give more explanation or context about the proposed method.

**Essential References Not Discussed:**

I am not an expert in the specific area of active sampling for causal inference. Therefore, I am not sure whether any important reference is missed.

**Experimental Designs Or Analyses:**

I did not check the codes, but I read the experiment section. Based on the text, it seems that the experimental results are sound and the authors also provided some explanations for the plots.

**Methods And Evaluation Criteria:**

The main metric for comparing the methods is a sample query to achieve an $\epsilon$-approximation error which is a reasonable evaluation criterion.

**Other Comments Or Suggestions:**

I suggest revising the whole paper and making sure that all the terms are defined. Moreover, please provide a general description of the proposed method first and then describe the details of each part. In the current version, for instance, it is hard to understand how Algorithm 1 works such as what is the purpose of GSW design. As another example, the term HT is used before saying that it refers to Horvitz-Thompson estimator.

**Other Strengths And Weaknesses:**

Strengths:
- The proposed method provides an order optimal method for active sampling to estimate the causal effect. For this purpose, they also gave a lower bound on the number of queries of any algorithm to achieve an $\epsilon$-approximation error.

- The experimental results showed that the proposed method has better performance compared to previous work in most real datasets.

Weakness:
- Some parts of the paper is not well written and some notations or terms are used without defining them which make hard to read the paper or validate the proofs.

**Questions For Authors:**

- What are technical novelties compared to Chen & Price (2019)? It seems that the results are adoptation of results in Chen & Price (2019) to the current setting. The explanation of the proposed method in lines 134-142 is vague. For instance, it is not clear "This approach serves as a refinement of Chen & Price (2019), in which we transfer the strategy (Definition 5 in Chen & Price (2019)) to the design-based setting."
- Please explain Algorithm 1 line by line and explain the intuition behind each part there.
-  The lines 149-157 are not written well. For instance, "We say it is a “good $\epsilon$-reweighting sampling strategy” if (i) Defining a matrix". Moreover, what is the second condition (ii) here?
- Regarding the statement of Theorem 5.1, first please fix "$\forall$ any" to "For any". Moreover, the statement is a little bit weird as it says that "for any algorithm whose output $\hat{\tau}$ satisfying $|\hat{\tau}-\tau|\leq 0.1$ with probability 0.75", then sample queries is at least $2.86d/\epsilon$ but $|\hat{\tau}-\tau|\leq 0.1$ does not depend on $\epsilon$ and it is just needed to be less than 0.1.
- In line 287, what is the notation "*"? In line, what is the notation $\tilde{\mathcal{N}}(i)$?

**Relation To Broader Scientific Literature:**

Identifying causal effects is one of the main goals in many areas of empirical science. One of the main approaches in causal effect identification is to perform experiments. This paper considers active sampling in experiment design in order to provide order-optimal methods in estimating the causal effect.

**Theoretical Claims:**

Unfortunately, I only had time to check the proof of Lemma 4.2.

---

> ### Author Rebuttal · Authors · 2025-04-01
>
> Thank you for your thoughtful review and valuable questions! We address your questions point-to-point in the following.
>
> > **Q1: However, the authors should provide more explanation or context about the proposed method.**
>
> 1. **Insightful motivation**.
>
> **Motivating example**. Suppose $X$ is $n \times 2$, and we want a subset to approximate $X^\top X$ for regression. Random sampling may cluster points along a single direction, leading to rank deficiency or poor conditioning. In contrast, Algorithm 2 adapts sampling probabilities to promote diversity: it increases the chance of selecting points in underrepresented directions by updating $B_i$. This ensures that new samples help span missing dimensions. After a few rounds, the selected subset captures both primary axes and preserves the geometry.
>
> To avoid over-representing outliers, constraints on $\alpha_i$ limit how much influence any point can have. As a result, the spectrum of the reweighted matrix $A^\top A$ remains close to that of the full dataset (identity matrix after normalisation). We avoid redundancy while preserving rare but critical directions—achieving near-optimal regression performance with far fewer, well-chosen samples.
>
>
>
> 2. **Detailed interpretation of main algorithm**
>
> | **Step**     | **Description** |
> |----------------------|-----------------|
> | Line 1       | Subsequently assign the Bernoulli trial to the cases selected by IRD. |
> | Lines 2–4    | The IRD Algorithm (Alg. 2) is deployed to select a subset $S$ representative to the covariate distribution of the entire samples and derive the estimation on adjustment parameter $\hat{\beta}^{act}$. |
> | Line 5       | Deduce the Horvitz-Thompson estimator on $S$ following the Bernoulli trial assigned in line 1. |
> | Lines 6–7    | Randomly sample a subset from the complementary set of $S$ and deploy the GSW design to derive a balanced assignment. Adjust the HT estimator to improve efficiency. |
> | Line 8       | Weight the causal effect estimators on representative set $S$ and set $\bar{S}_{m'}$ to yield the combined estimator using well-balanced assignments and well-representative samples. |
>
> Another line of motivation, refer to `RW RneH` for the **Q4**.
>
>
>
>
> 3. **Additional comparision**
> * **The technical novelty (i)**: First, we refine the previous result in Chen & Price (2019) to adapt to a finite-sample setting. It resorts to the refinement of the parameter in the $\epsilon$-reweighting strategy, compared to that of Chen & Price (2019), Definition 2.1. Intuitively, consider the OLS estimator regressing $Y$ on $X$ for any arbitrary weight $w_i$. We have
> $E[w_i (\hat{Y}_i^{ols} - Y_i)] = 0$, so the expectation of the noise term vanishes. However, in finite samples, such sum may usually deviate from zero, necessitating further refinement of the weighting parameters.
>
> * **The technical novelty (ii)**: It brings significant challenges since, compared with the setting in Chen & Price (2019), each individual's outcome is not fixed as $Y_i$, instead, the outcome is randomly chosen as $Y_i(0), Y_i(1)$, jumping between the real world and the counterfactual world.
>
> > **Q2: technical novelties compares to Chen & Price (2019)?**
>
> Kindly see **Q1-Additional comparison**
>
> > **Q3: Explain Algorithm 1**
>
> Kindly see **Q1-Detailed interpretation of main algorithm**.
>
> >**Q4: The lines 149-157's writing, and the meaning of Condition (ii).**
>
>
> The second condition (ii) is intended to impose an upper bound on the total weighting factors $\alpha_i$. Specifically, this condition ensures that the cumulative weights are controlled, preventing any single sample or a small group of samples from excessively dominating the overall estimator.
>
> To clarify the mathematical intuition, we provide the following rewritten version of the paragraph. Briefly, it includes (i) Spectral approximation condition, (ii) Bounded cumulative weight condition and (iii) Single-point influence condition. Due to space limitations, we kindly refer you to the anonymous link: https://anonymous.4open.science/r/Causal-Active-Sample-B0C5/README.md.
>
> > **Q5: The parameter analysis of the lower bound**
>
> Overall, whether setting it as $|\hat{\tau}-\tau|\leq \epsilon$ or $0.1$ are both correct (as long as $\epsilon < 0.1$); We sincerely appreciate your suggestion—explicitly including the $\epsilon$ parameter in the estimation error bound improves clarity and helps readers better understand the overall process and how the Shannon-Hartley upper bound applies to the estimation error.
>
> > **Q6: Notation meaning**
>
> “∗” is a “null option”; also refer to the dear reviewers to Algorithm 4 in the appendix. the notation $\mathcal{N}_b^{\pi}(i)$ is deferred in the above anonymous link.
> ***
> We are eager to hear your feedback. We’d deeply appreciate it if you could let us know whether your concerns have been addressed.

---

> > ### Comment · Reviewer_PucS · 2025-04-02
> >
> > Thanks for the response. It is now somewhat clearer to me what the contributions of the current work are. However, I still believe that the paper is not well-written — at least, I personally find it difficult to follow. This might be due to the fact that I am not a complete expert in this specific area of causal inference, and some concepts that the authors assume to be known could benefit from clearer explanations. To fairly assess this paper, I suggest relying more on the other reviewers who appear to be more familiar with this area.
> >
> > Regarding the responses, I still have one question about the lower bound. I think the statement in the paper, as well as the explanation in the rebuttal, is not quite correct. If I choose a very small $\epsilon < 0.1$, then an estimator that merely guarantees $|\tau - \hat{\tau}| < 0.1$ would still require access to $\Omega(d/\epsilon)$, which is quite counterintuitive. I still believe that the condition $|\tau - \hat{\tau}| < 0.1$ should be replaced with $|\tau - \hat{\tau}| < \epsilon$, and that the authors are actually proving this bound for any $\epsilon \in (0, 0.1)$.

---

> > > ### Author Response · Authors · 2025-04-02
> > >
> > > Thank you very much for your prompt response!
> > >
> > > ____
> > >
> > > Such a counterintuitive case of **data generation** does exist—where the estimation error is only required to be controlled within $0.1$ with some probability, yet achieving this still objectively requires samples of at least order $O(d/\epsilon)$.
> > >
> > > **Intuition** Even under the mild condition of the estimation error, researchers can still generate "hard, unsatisfactory" instances to ensure that its estimation is "difficult", requiring many samples.
> > >
> > >
> > > **Example** In our Appendix E ("The proof of the lower bound"), we provide a counterintuitive construction of data generation under the super-population perspective: We consider the following constructions: $\tau\left({X}_i\right)=L\left({X}_i\right)+\mu$. Here $L(\cdot)$ is the pre-fixed function, selected from the linear family $\mathcal{L}$ satisfying $\parallel \mathcal{L}\parallel_D=1$. $\mu$ is "important", namely, the i.i.d. Gaussian noise satisfying $\mu \sim N\left(0, \frac{1}{\varepsilon}\right) . D$ is a uniform distribution under the $d$ dimensional Euclidean space. We consider that ${X}_i$ is sampled from $D$. We set the  $\tau(X_i)$ as the corresponding ITE value $\tau_i$, and then we can naturally provide a feasible construction of $Y_i(1), Y_i(0)$.
> > >
> > > **Why this example makes sense.**
> > > We follow the technique in the active sampling.
> > >
> > > **Step 1** **First, there exists a subset {$\mathcal{L}=\{L_1, L_2,...L_s\}$} which satisfies $s \geq 2^{1.919d}$ and each pair among them contains a nontrivial "distance".** Specifically, $\exists$ a subset $\{L_1, L_2,... L_s\}=: \mathscr{L} \subseteq \mathcal{L}$ with $s \geq 2^{0.7d}$, $\parallel L_i\parallel_D = 1$, $\parallel L_i \parallel_{\infty} \leq 1$. Moreover, $ \parallel L_i-L_j \parallel_D \geq 0.2$.
> > >
> > >
> > > > This construction is achieved via a recursive greedy algorithm. As we illustrated in Appendix E (page 17), we first restrict $\mathcal{L}$ to all function mappings to ${\{ \pm 1 \} }^d$. On this basis, we recursively pick the legitimate function from it and in each step guarantees that we remove the function within the distance $0.2$ from our selected one. This process will output at least $2^{1.919d}$ functions according to our Lemma E.1, Appendix 4.
> > >
> > >
> > > **Step 2** **We make a lower bound of the mutual information between the randomly chosen generation function $L_j$ and any algorithm's output.**
> > >
> > >
> > > > Let $I(L_j; \hat{\tau})$ denote the mutual information of a uniformly randomly chosen function $L_j \in \mathcal{L}$ and algorithm's output $\hat{\tau}$ given $|\mathcal{L}|=s$ observations $(X_1,...), (X_2,...), ...(X_s, ...)$ generating from $\tau\left({X}_i\right)=L_j\left({X}_i\right)+noise\sim \mathcal{N}(0,\frac{1}{\epsilon})$. By Fano's inequality, we get
> > > $I(L_j; \hat{\tau}) = H(L_j) - H(L_j \mid \hat{\tau}) \geq 0.4log(s) \geq 1.43$ ($H(\cdot)$ is an entropy).
> > >
> > >
> > >
> > >
> > >
> > > **Step 3** **Third, we can also get an upper bound of the above mutual information.**
> > >
> > > > By Shannon–Hartley theorem, it leads to $I(L_j; \hat{\tau}) \leq \frac{s \epsilon}{2}$.
> > >
> > > Combined with **Steps 1-3**, it leads to the expected result, i.e, $1.43 \leq I(L_j; \hat{\tau}) \leq \frac{s \epsilon}{2}$ leads to $s = \Omega(2.86d /\epsilon)$, namely, the requirement of $O(d/\epsilon)$ samples.
> > >
> > >
> > >
> > >
> > >
> > > Inspired by for insightful consideration, we polish the statement of our theorem (under superpopulation), and will be presented in the Camera-ready version:
> > >
> > > > Theorem 5.1. (Lower bound). $\forall$ any fixed dimension $d$, for any $\epsilon \in(0,0.1)$ sufficiently small, there exists a feasible set of $\{\boldsymbol{Y}^1, \boldsymbol{Y}^0\}$ such that for any algorithm whose output $\hat{\tau}$ satisfying $|\hat{\tau}-\tau| \leq 0.1$ with probability $0.75$ , need at least $2.86 d / \varepsilon$ sample queries. Specifically, such data generation process could be chosen as i.i.d Gaussian noise $\mu = N(0, \frac{1}{\epsilon})$, and $\tau\left({X}_i\right)=L\left({X}_i\right)+\mu$, where $X_i$ sampled from some distribution $D$, and $\parallel L \parallel_D = 1$.
> > >
> > >
> > > **Insight** It demonstrates that the sample complexity of our method is optimal and cannot be improved. This is because, even under a relaxed requirement on the estimation error, there still exist challenging instances for which achieving accurate estimation requires a complexity of $O(d / \epsilon )$.
> > >
> > > ____
> > >
> > > References
> > >
> > > T. Cover, J. Thomas (1991). Elements of Information Theory. pp. 38–42. ISBN 978-0-471-06259-2.
> > >
> > >  Hartley, R. V. L. (July 1928). "Transmission of Information" (PDF). Bell System Technical Journal. 7 (3): 535–563.
> > >
> > >
> > > ____
> > >
> > > Your suggestions have already been incorporated into the revised version to improve its clarity and accessibility for a broader research audience. Moreover, it would mean a great deal if our efforts to address your concerns could be reflected in your evaluation. We would be deeply grateful if you would consider revisiting your initial score in light of our discussions with our joint effort.

---

### Official Review · Reviewer_kg3L · 2025-03-17

**Overall Recommendation:** 4

**Summary:**

This paper developed a finite-sample estimator with sample complexity analysis for causal effect estimation. The paper demonstrated the near-optimality of the sample size, and further extended the framework to social networks. Numerical experiments with simulated and real-world data supported the effectiveness of the proposed estimator.


### update after rebuttal:
I have read through the authors' response and will be keeping my original scores.

**Claims And Evidence:**

Overall the proposed framework is well-described and the main claims in the paper are well-supported. The main theoretical results include the ATE estimation error upper bound and a matching lower bound under a superpopulation perspective. The proof sketch provided for the upper bound is clearly presented.

**Essential References Not Discussed:**

I did not notice missing essential references.

**Experimental Designs Or Analyses:**

Based on the presented content in the main paper, the experimental design appears to be sound.

**Methods And Evaluation Criteria:**

The upper bound was established via two main steps: the IRD algorithm adapted from Chen & Price (2019)'s design-based setting results, and using the obtained coefficients from IRD to adjust the finite-sample estimator. The lower bound shows that this upper bound is near-optimal. Based on the experiments with synthetic and real-world data, the proposed estimator did not substantially outperform all baselines, but achieved comparable or better results in most of the scenarios.

**Other Comments Or Suggestions:**

Minor typo: section 4 1st paragraph, Iterative Reweighting in Design-based "Set- ting"

**Other Strengths And Weaknesses:**

The paper was well-organized, claims and proof sketches were clearly stated.

**Questions For Authors:**

In the experiments, there are several settings where CRA / GSW actually outperformed the proposed estimator. Can you provide more intuition or justifications when the proposed estimators tend to perform better or worse?

**Relation To Broader Scientific Literature:**

The broader literature was well-addressed.

**Theoretical Claims:**

To my best knowledge the upper bound proof is correct.

---

> ### Author Rebuttal · Authors · 2025-04-01
>
> ### **Acknowledgement and General Response to `RW kg3L`, `RW PucS` , `RW RneH`, `RW fVjN`** ###
> Thrilled to receive such a positive reception from dear reviewers! Also, sincerely thank all reviewers for their insightful suggestions! In this general response, we carefully synthesized the reviewers’ suggestions to facilitate the AC’s assessment and, more importantly, use them as a blueprint to guarantee the quality of the camera-ready version. We have addressed these points:
>
> 1. **Detailed and intuitive explanation of motivation, methods and algorithms** (`RW PucS`, `RW RneH`). In our rebuttal and our revised version, we provide _**(i) insightful motivation and intuition, (ii) additional comparison with previous literature and (iii) detailed interpretation of the main algorithm**_, especially for audiences who are unfamiliar with causality or active sampling.
>
> 2. **Experiments** (`RW kg3L`, `RW fVjN`).  We provide additional performance comparisons upon the dimension $d$, the size $n$, the network interference, etc., to check in which circumstance the method is more advantageous. These empirical results back up our theories and intuitions.
>
> 3. **Mention some misleading statements and notations** (`RW kg3L`, `RW PucS`, `RW fVjN`). Thanks for the meticulousness. Our improved version has made a few isolated definitions more transparent and intuitive.
>
> If have any further questions, please don’t hesitate to contact us directly. We will respond as promptly as possible and do our utmost to address every concern thoroughly. Let’s enjoy this fruitful discussion together.
>
> ----
> ### **For `RWkg3L`: Thanks and please kindly find below our concise and clear rebuttal addressing your concerns.** ###
> Truly appreciate your strong endorsement! Here is our response:
> > **Q1: the synthetic experimental design appears to be sound. In the experiments, there are several settings where CRA / GSW actually outperformed the proposed estimator. Can you provide more intuition or justifications as to why the proposed estimators tend to perform better or worse?**
>
> Thanks for your advice! More intuitions are as follows:
> 1. **The advantage of our RWAS algorithm over traditional CRA/GSW methods is especially larger when the sample dimension $d$ is large**. This is because we have proven, through bounding arguments, that the sample complexity of RWAS is relatively optimal during the active sampling process that achieves a (1+\epsilon) relative fitting error, which is O(d/\epsilon). As a result, when the sample dimension dd is relatively large, our error is minorer than that of other methods.
>
> 2. **The data structure (i.e., whether there exists a subset with efficient global representational capability) plays an important role in comparing the performance of these algorithms**. Compared to the GSW algorithm, our key differences are that
>     * We adopt the IRD active sampling strategy (yielding a subset whose $X^\top X$ distribution closely mirrors the overall distribution), thus obtaining regression coefficients from $X$ to the individual treatment effect (ITE = Y(1)-Y(0)); and
>     * We apply regression adjustment to GSW itself (leveraging the regression coefficients from the former step).
>  Intuitively, suppose the covariate distribution of this subset closely approximates the overall distribution. In that case, (i) our regression-adjusted GSW will be more efficient, and (ii) our IRD will produce a smaller fitting error for estimating the overall ATE.
>
> 3. **Motivated by your inspiration, we provide the following supplementary experiments** on the performance differences as d varies, which aligns with the above intuition.
>
> | d     | HT         | Hajek      | CRA         | GSW         | RAHT        | SC          | 4-Vectors   | RWAS (Ours)   |
> |-------|------------|------------|-------------|-------------|-------------|-------------|-------------|---------------|
> | 10    | 1.98 (0.78) | 1.27 (0.53) | 0.98 (0.40) | **0.93 (0.38)** | 1.08 (0.47) | 1.13 (0.65) | 1.15 (0.53) | 1.08 (0.46)    |
> | 20    | 3.67 (0.88) | 2.88 (0.57) | **2.10 (0.49)** | 2.28 (0.70) | 2.41 (0.59) | 2.50 (0.73) | 2.46 (0.62) | 2.19 (0.55)    |
> | 50    | 1.14 (0.74) | 1.03 (0.50) | 1.02 (0.39) | 0.80 (0.64) | 0.80 (0.56) | 0.80 (0.68) | 0.92 (0.61) | **0.71 (0.47)** |
> | 100   | 1.82 (0.95) | 1.62 (0.86) | 2.22 (0.53) | 1.78 (0.87) | 1.53 (0.83) | 1.73 (0.90) | 1.80 (0.85) | **1.51 (0.82)** |
>
> The results indicate that our RWAS methods dominate the previous methods, especially when $d$ is large, which is relevant in practice. The code is available at https://anonymous.4open.science/r/Causal-Active-Sample-B0C5/README.md.
>
> > **Q2: Minor typo.**
>
> Sincerely, thanks for your careful review! We revise this topo from "Set-ting" to "Setting".

---

### Decision · Program_Chairs · 2025-05-01

**Decision:**

Accept (poster)

**Comment:**

Almost all reviewers are favorable to the paper.

There is one issue with the lower bound with the way it was stated. Authors show that a specific data generating process could explain the counterintuitive-looking claim in the lower bound (I read through it and seems satisfactory). I encourage the authors to update the statement of the theorem for lower bound as they have promised.

Authors show that given n units with covariates to be assigned for treatment and control group, in the active learning setting authors propose a way to sub select only $O(d/epsilon)$ units for assignment and also propose a way to partition them also to achieve an ATE estimation error of at most $epsilon$ when the potential outcomes follows a linear model. Authors exploit a sub selection procedure of Chen and Price from 2019 that actively sub selects covariates for labeling for linear regression that is representative of the full set of covariates and further combining it with partitioning using Gram-Schmidt Walk type experimental design of Harshaw et al 2024 that preserves covariate balancing.

Given the results of Chen and Price 2019 and its advantages over leverage score sampling (employed in previous ATE active estimation works), the results here are not totally surprising. However, the final result establishes near optimal sample complexity tightening existing results.

Therefore I recommend accept.